# Discovery of coordinately regulated pathways that provide innate protection against interbacterial antagonism

See-Yeun Ting[1], Kaitlyn D LaCourse[1], Hannah E Ledvina[1], Rutan Zhang[2], Matthew C Radey[1], Hemantha D Kulasekara[1], Rahul Somavanshi[1], Savannah K Bertolli[1], Larry A Gallagher[1], Jennifer Kim[1], Kelsi M Penewit[3], Stephen J Salipante[3], Libin Xu[2], S Brook Peterson[1], Joseph D Mougous[1,4,5]*

[1]Department of Microbiology, University of Washington School of Medicine, Seattle, United States; [2]Department of Medicinal Chemistry, University of Washington School of Pharmacy, Seattle, United States; [3]Department of Laboratory Medicine and Pathology, University of Washington School of Medicine, Seattle, United States; [4]Department of Biochemistry, University of Washington School of Medicine, Seattle, United States; [5]Howard Hughes Medical Institute, University of Washington, Seattle, United States

*For correspondence:
mougous@uw.edu

**Abstract** Bacterial survival is fraught with antagonism, including that deriving from viruses and competing bacterial cells. It is now appreciated that bacteria mount complex antiviral responses; however, whether a coordinated defense against bacterial threats is undertaken is not well understood. Previously, we showed that *Pseudomonas aeruginosa* possess a danger-sensing pathway that is a critical fitness determinant during competition against other bacteria. Here, we conducted genome-wide screens in *P. aeruginosa* that reveal three conserved and widespread interbacterial antagonism resistance clusters (*arc1-3*). We find that although *arc1-3* are coordinately activated by the Gac/Rsm danger-sensing system, they function independently and provide idiosyncratic defense capabilities, distinguishing them from general stress response pathways. Our findings demonstrate that Arc3 family proteins provide specific protection against phospholipase toxins by preventing the accumulation of lysophospholipids in a manner distinct from previously characterized membrane repair systems. These findings liken the response of *P. aeruginosa* to bacterial threats to that of eukaryotic innate immunity, wherein threat detection leads to the activation of specialized defense systems.

## Editor's evaluation

This study identifies novel (Arc) pathways that mediate resistance of bacteria to attacks by other bacteria. The identified genes appear to specifically protect host cells from the action of bacterial toxins. In particular, one system (Arc3) is shown to protect *Pseudomonas aeruginosa* from Type-6 secretion phospholipase effectors. Thus, while it is growing increasingly clear that bacteria have evolved numerous protection mechanisms against phages, the study expands this concept to bacteria.

## Introduction

Antagonism from other organisms is a threat faced nearly universally by bacteria living in mixed populations, yet we are only beginning to understand the mechanisms employed in defense against these

assaults (*Hersch et al., 2020a*; *Peterson et al., 2020*; *Robitaille et al., 2021*). One defense mechanism that is common in the gut microbiome and potentially other habitats is the production of immunity proteins that grant protection against specific toxins delivered by the type VI secretion system (T6SS) (*Ross et al., 2019*). These 'orphan' immunity proteins share homology with and are likely evolved from cognate immunity proteins that protect bacteria from undergoing self-intoxication, which is inherent to the indiscriminate nature of the T6SS delivery mechanism. Core cellular structures have also been implicated in promoting survival during interbacterial antagonism. Extracellular polysaccharide capsules protect *Vibrio cholerae* and *Escherichia coli* from T6SS-based attack of other species (*Hersch et al., 2020b*; *Toska et al., 2018*), and in *Acinetobacter baumannii*, lysine modification at peptidoglycan crosslinks was suggested to render the cell wall resistant to degradation by amidase toxins (*Le et al., 2020*). Finally, characterized stress response pathways also appear to contribute to antagonism defense. Indeed, it has been suggested that bacterial stress responses evolved in part as a means of detecting and responding to antagonistic competitors (*Cornforth and Foster, 2013*; *Lories et al., 2020*). A candidate-based approach applied to *E. coli* and *P. aeruginosa* found that stress response genes involved in envelope integrity and acid stress, among others, can contribute to resistance against a *V. cholerae* phospholipase toxin (*Kamal et al., 2020*). While the *P. aeruginosa* genes identified impacted survival when the toxin was produced heterologously within the organism, they did not influence the fitness of the bacterium during interbacterial competition with *V. cholerae*.

We previously discovered that *P. aeruginosa* mounts an effective but largely undefined defense upon exposure to an antagonistic competitor, which we named the **P. a**eruginosa **r**esponse to antagonism (PARA) (*LeRoux et al., 2015a*). The response is activated when a subpopulation of *P. aeruginosa* cells are lysed, releasing intracellular contents that trigger the response in neighboring survivors (*LeRoux et al., 2015b*). PARA is coordinated by the Gac/Rsm global regulatory pathway, which post-transcriptionally controls the expression of nearly 400 genes (*Goodman et al., 2004*; *Lapouge et al., 2008*). This regulon includes the Hcp secretion island I-encoded type VI secretion system (H1-T6SS), the activity of which enables *P. aeruginosa* to kill or disable competing bacteria through the delivery of a cocktail of toxic effector proteins (*Hood et al., 2010*). While an important component of PARA, data suggest that the H1-T6SS is just one feature of the response; *P. aeruginosa* strains lacking functionality of the two-component system required to initiate Gac/Rsm signaling are severely crippled in interbacterial antagonism defense relative to those lacking only the H1-T6SS (*LeRoux et al., 2015a*). The majority of the genes under Gac/Rsm control encode proteins of unknown function, and we hypothesized that these could represent uncharacterized, novel mechanisms by which *P. aeruginosa* defends against interbacterial antagonism.

## Results

### Discovery of coordinately regulated *P. aeruginosa* interbacterial toxin defense pathways

Prior work by our laboratory has shown that *Burkholderia thailandensis* (*B. thai*) fiercely antagonizes *P. aeruginosa* using its antibacterial T6SS (*LeRoux et al., 2015a*). Thus, to identify its interbacterial defense factors, we subjected a transposon library of *P. aeruginosa* to antagonism by *B. thai* or a derivative lacking antibacterial T6SS activity (ΔT6S) and used high-throughput sequencing to measure gene-level fitness changes. Duplicate screening of this library revealed 34 genes critical for the survival of *P. aeruginosa* specifically while undergoing antagonism by *B. thai* (*Figure 1A*, *Figure 1—figure supplement 1A and B* , *Figure 1—source data 1*). Strikingly, 25 of these belong to the Gac/Rsm regulon and three additional hits included genes directly involved in the Gac/Rsm signaling system (*Goodman et al., 2004*; *Lapouge et al., 2008*). Indeed, disruption of genes encoding GacS and GacA, the central sensor kinase and response regulator required for activation of the pathway, respectively, elicited the strongest interbacterial defense defect among all genes in the *P. aeruginosa* genome (*Figure 1A and B*). On the contrary, inactivating insertions in *retS*, encoding a hybrid sensor kinase that negatively regulates GacS, increased interbacterial competitiveness.

Our screens identified previously uncharacterized genes belonging to three operons under Gac/Rsm control (*Goodman et al., 2004*), several genes within two operons of the HSI-I-encoded T6SS (H1-T6SS) (*Hood et al., 2010*), and two genes within an operon not subject to Gac/Rsm regulation (*Figure 1C*, *Figure 1—figure supplement 1C*, *Figure 1—source data 1*). Given that false positives

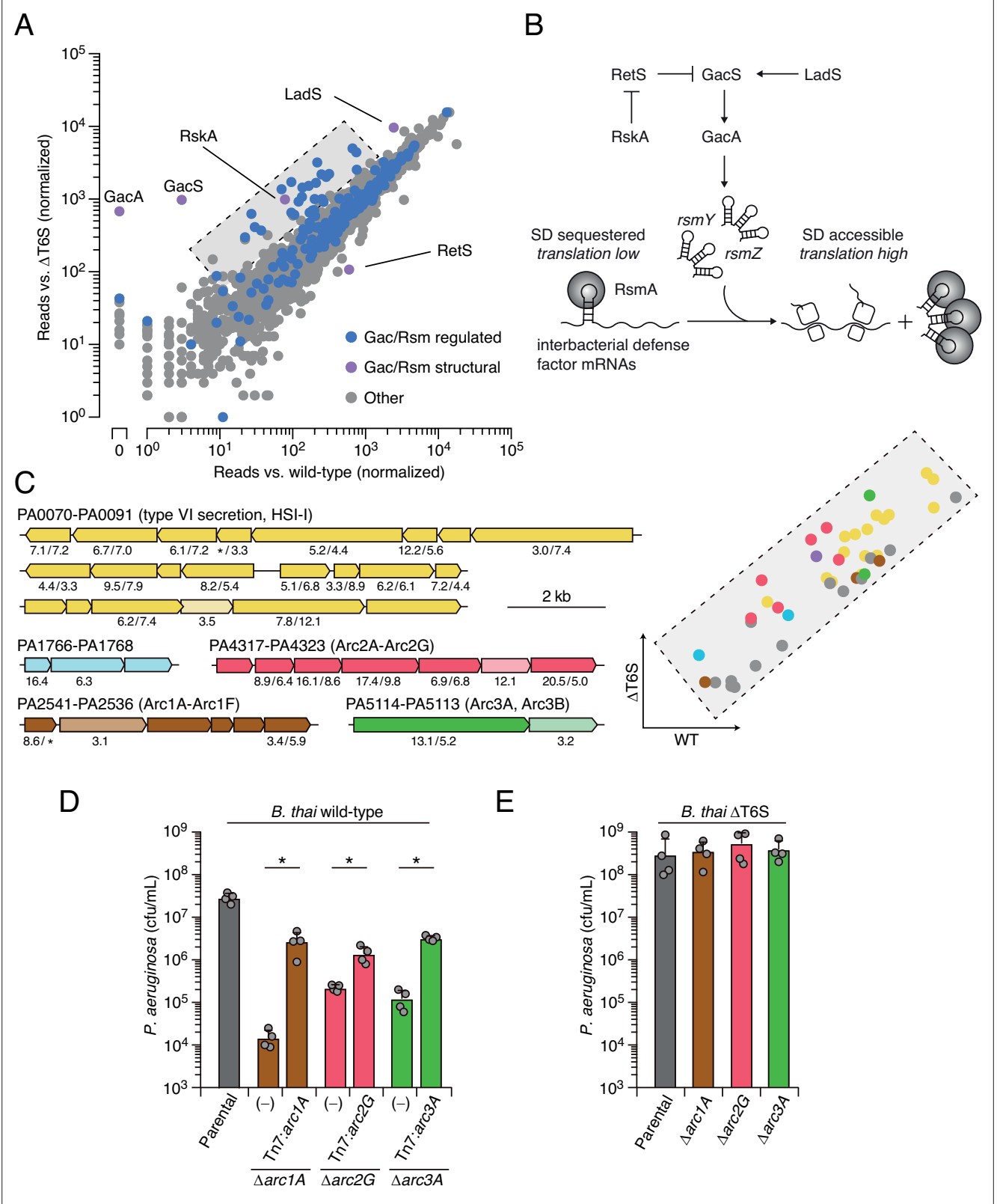

**Figure 1.** Multiple pathways under Gac/Rsm control contribute to *P. aeruginosa* defense against antagonism. (**A**) Transposon library sequencing-based comparison of the fitness contribution of individual *P. aeruginosa* genes during growth competition with wild-type *B. thai* versus *B. thai* ΔT6S. Genes under Gac/Rsm control (blue) and those encoding core Gac/Rsm regulatory factors (purple, labeled) are indicated. (**B**) Overview of the Gac/Rsm pathway (SD, Shine–Dalgarno). The *rsmY* and *rsmZ* genes encode small RNA molecules that sequester the translational regulator RsmA. Hybrid

*Figure 1 continued on next page*

*Figure 1 continued*

sensor kinase PA1611 is here renamed RetS-regulating sensor kinase A (RskA). (**C**) Left: *P. aeruginosa* gene clusters hit (greater than threefold in replicate screens; *Figure 1—source data 1*) in this study. Numbers below genes indicate transposon insertion ratio (*B. thai* ΔT6S/wild-type) for screen in (**A**) followed by the replicate screen (*Figure 1—figure supplement 1*); light toned genes were hit in only one replicate. Asterisk indicates genes for which insertion was undetected in libraries obtained from *B. thai* wild-type competition. Right: zoom-in of boxed region of (**A**) with genes colored corresponding to clusters at left. (**D, E**) Recovery of *P. aeruginosa* cells with the indicated genotypes following growth competition against *B. thai* wild-type (**D**) or ΔT6S (**E**). For interbacterial competition assays, the mean ± SD of biological duplicates and associated technical duplicates is shown: *p<0.05 using unpaired two-tailed Student's *t*-test.

The online version of this article includes the following source data and figure supplement(s) for figure 1:

**Source data 1.** Transposon sequencing-based analysis of *P. aeruginosa* fitness determinants during antagonism by *B. thai* compared to *B. thai* with an inactive T6SS (Δ*icmF*).

**Figure supplement 1.** Multiple pathways under Gac/Rsm control contribute to *P. aeruginosa* defense against antagonism.

**Figure supplement 2.** Antagonism resistance clusters possess functionally related genes and are in diverse bacteria.

**Figure supplement 3.** Interbacterial competition assays measure the contribution of *arc1-3* to *P. aeruginosa* fitness during antagonism by *B. thai.*

are less common among multiple gene hits within an operon, we focused on these genes for subsequent validation in pairwise competition assays. Strains bearing in-frame deletions of the genes most strongly hit in each operon showed substantially reduced fitness in competition with wild-type *B. thai*, and in each instance this could be partially restored by genetic complementation (*Figure 1D*). Further consistent with our screening results, each of the single-gene deletion mutants grew similarly to the wild-type in competition with *B. thai* unable to antagonize *P. aeruginosa* (*B. thai* ΔT6S) (*Figure 1E*).

## Arc1-3 operate independently and defend in a manner distinct from general stress responses

The *P. aeruginosa* Gac/Rsm pathway has been implicated in a process we termed danger sensing (*LeRoux et al., 2015a*; *LeRoux et al., 2015b*). We previously showed that the lysis of neighboring kin cells triggers the pathway, leading to T6SS activation. However, the Gac/Rsm pathway post-transcriptionally activates hundreds of genes whose function has largely remained cryptic (*Figure 1B*). Bioinformatic analyses suggested that each of the uncharacterized Gac/Rsm-regulated operons hit in our screen likely constitute a discrete pathway. Five of the six genes within the PA2536-41 cluster are predicted to encode proteins related to phospholipid biosynthesis or metabolism, including four genes encoding products with significant homology to enzymes required for phosphatidylglycerol synthesis (*Figure 1—figure supplement 2A*). Orthologs of core bacterial phospholipid biosynthetic machinery are encoded elsewhere in the genome, indicating likely specialization of this putative pathway. The PA4317-23 genes co-occur across many species within the Xanthomonadaceae and Pseudomonadaceae (*Figure 1—figure supplement 2B*). While none of the proteins encoded by this gene cluster have been characterized, they include a predicted MoxR-like AAA+-ATPase and a protein with a von Willebrand factor (VWF) domain. Related protein pairs co-occur widely and are known to cooperate in a chaperone-like function that releases protein inhibition or stimulates metal cofactor insertion (*Snider and Houry, 2006*; *Tsai et al., 2020*). Finally, PA5113 and PA5114 also co-occur widely, and several organisms encode single polypeptides that represent a fusion of the two proteins (*Figure 1—figure supplement 2C*).

Next, we sought to obtain experimental support for our hypothesis that the Gac/Rsm-regulated operons hit in our screen constitute independent functional units. First, we compared the fitness of our single-deletion strains to those bearing in-frame deletions in the two genes most strongly hit within each operon. In each operon, the second deletion did not impact competitiveness with *B. thai*, suggesting that the pairs of genes encode components of the same pathway (*Figure 1—figure supplement 3A*). On the contrary, additive effects on interbacterial competitiveness were observed when genes from each of the three operons were inactivated in a single strain. Indeed, the fitness deficiency of this triple mutant strain approaches that of the Δ*gacS* strain, which lacks Gac/Rsm function entirely (*Figure 1—figure supplement 3B*). Together, these data reveal that the Gac/Rsm pathway and discrete functional units under its control are critical for *P. aeruginosa* survival when antagonized by the T6SS of *B. thai*. Based on these data, we named the genes with the three Gac/Rsm-regulated

operons hit in our screen *arc1A-F* (**a**ntagonism **r**esistance **c**luster 1, PA2541-36), *arc2A-G* (PA4317-23), and *arc3A,B* (PA5114-13) (*Figure 1C*).

As a first step toward understanding how Arc1-3 influence the competitiveness of *P. aeruginosa*, we asked whether they serve defensive or offensive roles. In our initial screen, *B. thai* dramatically outnumbered *P. aeruginosa* (~50:1); therefore, the diminished *P. aeruginosa* viability observed under these conditions suggested each pathway can function defensively. However, these conditions do not exclusively isolate the defensive contribution of the pathways. For example, we observed that killing competitor cells via the H1-T6SS indirectly contributes to *P. aeruginosa* viability under these conditions, likely by expanding the available growth niche. To probe the capacity of Arc1-3 to antagonize *B. thai*, we initiated co-culture growth competition assays with an excess of *P. aeruginosa*. We previously demonstrated that in co-cultures initially dominated by *P. aeruginosa,* antagonism by *B. thai* has a negligible impact on *P. aeruginosa* growth (*Schwarz et al., 2010*). Under these conditions, inactivating Arc2 and Arc3 had no impact on *B. thai* survival. Arc1 inactivation led to an increase in *B. thai* survival; however, this effect was modest relative to its impact on *P. aeruginosa* defense (*Figure 2A*). In contrast, inactivation of the H1-T6SS increased *B. thai* survival by approximately four orders of magnitude, eclipsing its impact on *P. aeruginosa* defense by ~100-fold. Together, these finding indicate that the contributions of Arc1-3 to *P. aeruginosa* competitiveness during antagonism arise predominantly from defensive mechanisms.

Our results suggested that the Gac/Rsm system functions analogously to innate immune sensors of eukaryotic organisms, wherein danger sensing activates the expression of independently functioning downstream effectors that defend the cell against specific threats (*Paludan et al., 2021*). By extension, this response should be distinct from that provoked by general stressors. To test this, we measured the contribution of Arc1-3 to *P. aeruginosa* survival in conditions known to induce general stress response systems, including heat shock, exposure to $H_2O_2$, and propagation in high-salinity media or media containing detergent (*Jørgensen et al., 1999*; *Munguia et al., 2017*). Unlike previously established stress-intolerant control strains (ΔrpoS and ΔvacJ), the survival of strains lacking Arc1-3 function was equivalent to wild-type *P. aeruginosa* in these assays, suggesting that their function is distinct from those involved in general stress resistance (*Figure 2—figure supplement 1*). This is also in line with the regulatory profile of Arc1-3; immunoblotting showed that basal expression of each is exceedingly low, yet highly inducible through stimulation of the antagonism-responsive Gac/Rsm pathway (*Figure 2—figure supplement 2*).

## Arc1-3 provide defense against distinct threats

The specific involvement of Arc1-3 in defense against antagonism led us to hypothesize that the pathways could grant protection against mechanistically distinct threats. Notably, *B. thai* delivers a cocktail of toxins to target cells including two predicted phospholipases (Tle1 and Tle3) (*Russell et al., 2013*), a peptidoglycan-degrading amidase (Tae2) and a colicin-like toxin predicted to form inner membrane pores (ColA) (*Russell et al., 2012*; *Salomon et al., 2014*). To determine the role of Arc1-3 in defense against insults caused by toxins with biochemically diverse modes of action, we subjected strains with individual Arc pathways inactivated to pairwise competition against *B. thai* strains lacking single toxins in its arsenal. For Tle1 and Tle3, which are duplicated in the *B. thai* genome and required for basal T6SS function, strains bearing mutated codons corresponding to predicted catalytic residues of the proteins were utilized (*Figure 2—figure supplement 3*). Strikingly, we found that Tle3 inactivation abrogated the antagonism defense defect of *P. aeruginosa* lacking Arc3 function (*Figure 2B*), whereas its defense defect was maintained in competition experiments against other toxin-inactivated strains of *B. thai*. In contrast, the survival of *P. aeruginosa* lacking Arc2 function was significantly restored by deletion of *colA* in *B. thai*, while none of the single-effector mutations within *B. thai* impacted the defense defect of *P. aeruginosa* lacking Arc1 function. These results support the hypothesis that Arc pathways can afford protection in a manner specific to the damage arising from mechanistically distinct threats (Arc2, Arc3) or play a more general role in defense (Arc1). However, *B. thai* delivers toxins beyond those we inactivated in this study; therefore, we cannot rule out that Arc1 – like Arc2 and Arc3 – provides effector-specific defense.

We found that Arc2 and Arc3 are major defense factors against insults caused by ColA and Tle3, respectively, suggesting that these toxins may damage cells by a mechanism not efficiently countered by other cellular pathways. To evaluate the relative defensive contribution of Arc2 and Arc3

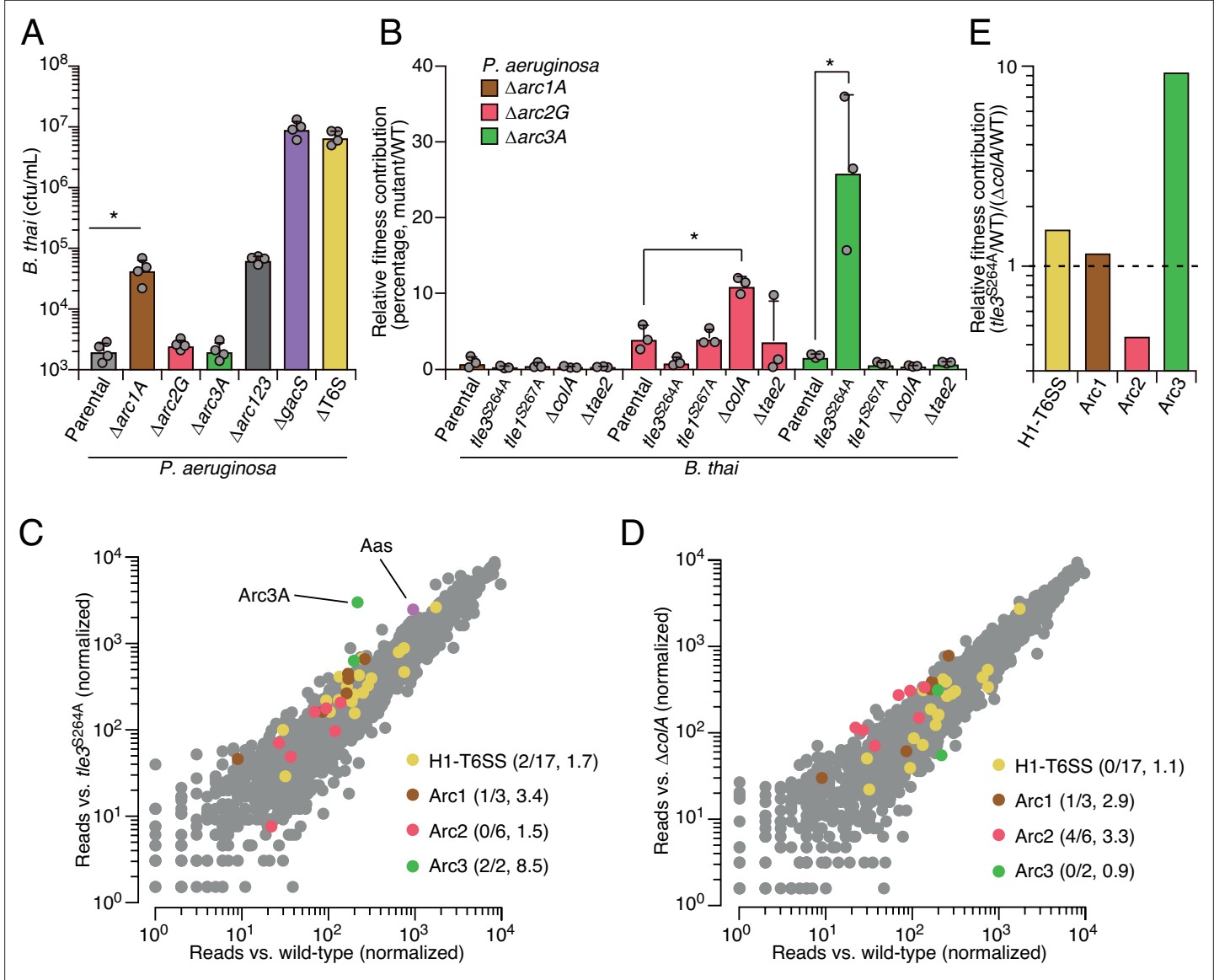

**Figure 2.** Arc pathways can provide toxin-specific antagonism defense that eclipses that of other cellular factors. (**A**) Recovery of *B. thai* cells following growth competition against a relative abundance of *P. aeruginosa* containing the indicated gene deletions, colored according to *Figure 1C*. (**B**) Relative survival of *P. aeruginosa* Arc1-3-inactivated mutants following growth in competition with an excess of the indicated *B. thai* strains (mutant CFU/parental CFU × 100 for each *B. thai* competitor). The loss of ColA or Tle3 activity in *B. thai* increases the relative survival of *P. aeruginosa* lacking Arc2 or Arc3 activity, respectively. (**C, D**) Transposon library sequencing-based comparison of the fitness contribution of individual *P. aeruginosa* genes during growth competition with wild-type *B. thai* versus *B. thai* lacking Tle3 (**C**) or ColA (**D**) activity, colored according to *Figure 1C*. Values in parentheses correspond to the number of genes hit (greater than threefold change in transposon insertion frequency) within each cluster compared to the number within each cluster hit in the initial screens (*B. thai* wild-type versus *B. thai* ΔT6S) followed by the average transposon insertion frequency ratio (*B. thai* mutant/*B. thai* wild-type) of the genes in the depicted screen. (**E**) Fitness contribution of each *P. aeruginosa* Arc pathway and the H1-T6SS to defense against Tle3 versus ColA. For interbacterial competition assays, the mean ± SD of biological duplicates and two technical replicates is shown: *$p < 0.05$ using unpaired two-tailed Student's *t*-test.

The online version of this article includes the following source data and figure supplement(s) for figure 2:

**Source data 1.** Transposon sequencing-based analysis of *P. aeruginosa* fitness determinants during antagonism by *B. thai* compared to *B. thai* tle3^S264A.

**Source data 2.** Transposon sequencing-based analysis of *P. aeruginosa* fitness determinants during antagonism by *B. thai* compared to *B. thai* ΔcolA.

**Figure supplement 1.** Arc1-3 do not contribute to the survival of *P. aeruginosa* exposed to common environmental stresses.

**Figure supplement 2.** Arc1-3 are subject to tight regulation by the Gac/Rsm signaling pathway.

**Figure supplement 2—source data 1.** Original uncropped image and original uncropped image with relevant bands indicated for *Figure 2—figure*

*Figure 2 continued on next page*

*Figure 2 continued*

**supplement 2**, Arc1A–V.

**Figure supplement 2—source data 2.** Original uncropped image and original uncropped image with relevant bands indicated for *Figure 2—figure supplement 2*, Arc1A–V RpoB control.

**Figure supplement 2—source data 3.** Original uncropped image and original uncropped image with relevant bands indicated for *Figure 2—figure supplement 2*, Arc2G–V.

**Figure supplement 2—source data 4.** Original uncropped image and original uncropped image with relevant bands indicated for *Figure 2—figure supplement 2*, Arc2G–V RpoB control.

**Figure supplement 2—source data 5.** Original uncropped image and original uncropped image with relevant bands indicated for *Figure 2—figure supplement 2*, Arc3A–V.

**Figure supplement 2—source data 6.** Original uncropped image and original uncropped image with relevant bands indicated for *Figure 2—figure supplement 2*, Arc3A–V RpoB control.

**Figure supplement 3.** The loci encoding Tle1 and Tle3 are duplicated in *B. thai,* and Tle3 induces potent self-intoxication.

genome-wide, we performed additional transposon-based screening in which a mutant library of *P. aeruginosa* was grown in competition with *B. thai* strains lacking ColA or Tle3 activity. Remarkably, this experiment defined *arc3A* as the gene most critical for *P. aeruginosa* survival during intoxication by Tle3 (*Figure 2C*, *Figure 2—source data 1*). Insertions in the second Arc3 gene, *arc3B*, also crippled *P. aeruginosa* defense against Tle3, though to a lesser degree. Arc2 genes were dispensable for Tle3 defense, and only one gene within Arc1 contributed to Tle3 defense beyond the threefold insertion frequency ratio cutoff we employed. Inactivation of ColA had a relative minor impact on the fitness of most *P. aeruginosa* transposon mutants (*Figure 2D*, *Figure 2—source data 2*). However, four of the six Arc2 genes hit in our initial screen provided defense against ColA that exceeded our threefold cutoff. Additionally, one Arc1 gene met these criteria, whereas Arc3 genes were wholly dispensable for defense against ColA. These results support the model that Arc3 provides specific defense against Tle3, Arc2 provides a degree of specific protection against ColA, and Arc1 serves a broader role in antagonism defense. A comparison of the fitness contributions of each Arc pathway and the H1-T6SS to *P. aeruginosa* undergoing intoxication by Tle3, ColA or Tle1 highlights the degree of their specificity (*Figure 2E*).

## Arc3 is a broadly conserved pathway that prevents lysophospholipid accumulation

The highly specific defense afforded by Arc3 against the predicted phospholipase Tle3 prompted us to further interrogate its function. Arc3A is a large protein containing 35 predicted transmembrane domains and lacking residues strongly indicative of enzymatic activity (*Figure 3A*). With predicted outer-membrane-anchored N-terminal lipidation and a C-terminal inner-membrane transmembrane helix, Arc3A is expected to span the periplasm. Neither Arc3A nor Arc3B share significant homology with characterized proteins; however, apparent orthologs of each are encoded by neighboring open-reading frames across an exceptionally wide distribution of bacteria (*Figure 3B*). Arc3A-related proteins are composed of domain of unknown function 2339 (DUF2339), possess large, yet varying numbers of transmembrane segments, and are found in bacteria belonging to most Gram-negative and Gram-positive phyla. Consistent with the predicted N-terminal localization of Arc3B to the outer membrane, its related proteins are restricted to Gram-negative phyla. Co-immunoprecipitation studies using epitope-tagged variants of Arc3A and Arc3B encoded at their native chromosomal loci provided evidence that the two proteins stably associate (*Appendix 1—table 1*). The *arc3A* and *arc3B* genes abut a third gene encoded in the same orientation, *estA,* and there is evidence that transcripts containing all three genes are generated by *P. aeruginosa* (*Gebhardt et al., 2020*). Although *estA* encodes an esterase (*Wilhelm et al., 1999*), which could have relevance for phospholipase defense, *estA* was not detected in our co-immunoprecipitation analyses, it did not exhibit differential insertion frequency in our screens, and in-frame deletion of *estA* did not impact *P. aeruginosa* survival during intoxication by Tle3 (*Figure 1—source data 1*, *Figure 2—source data 1 and 2*, *Figure 3—figure supplement 1A*).

The observation that Arc3A proteins are found in bacteria lacking an Arc3B homolog, combined with our finding that Arc3B contributes relatively little to Tle3 defense, led us to speculate that Arc3A

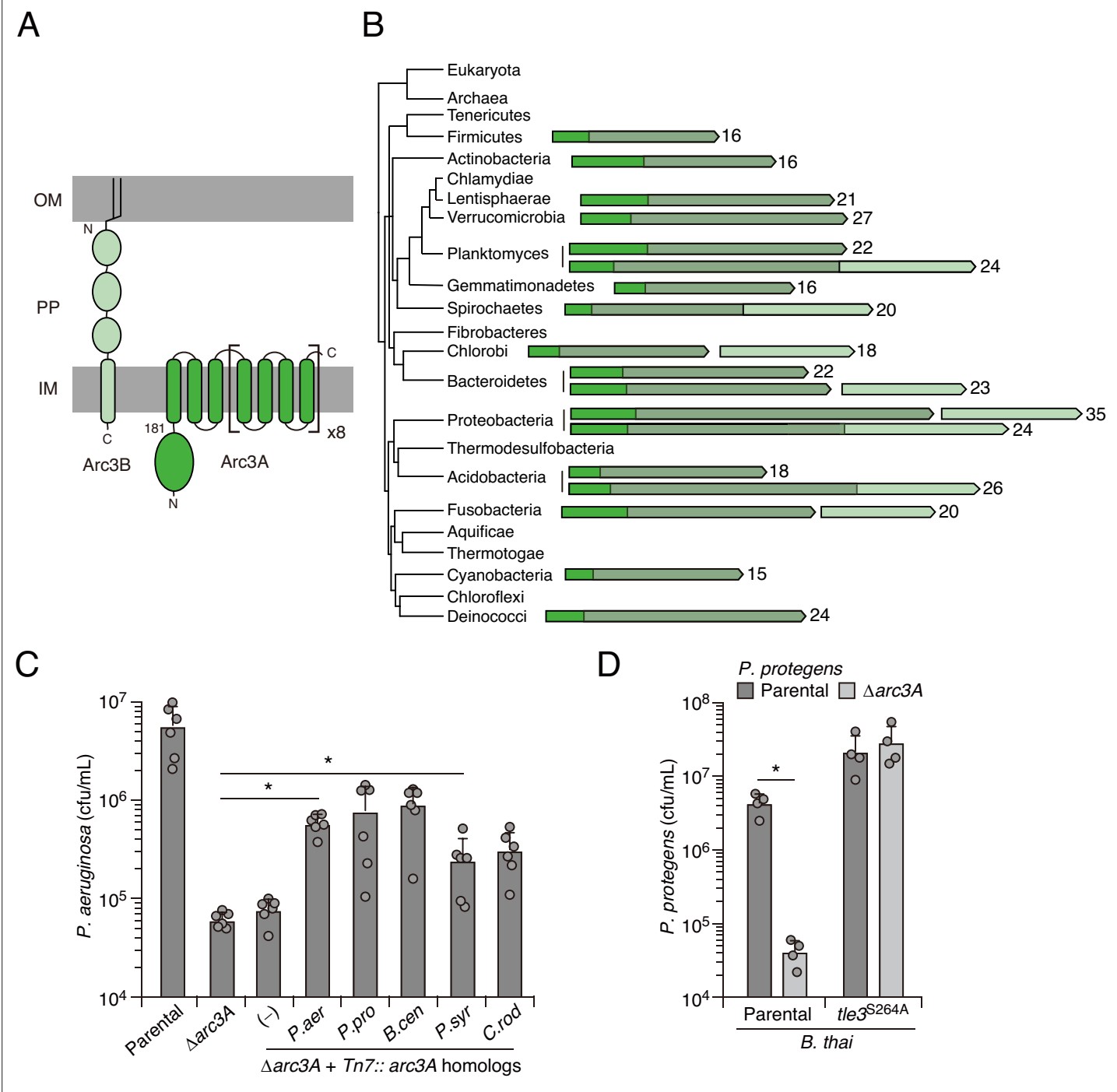

**Figure 3.** Arc3B is a predicted massive polytopic membrane protein with functionally complementing family members that occur widely in Gram-negative and -positive bacterial phyla. (**A**) Schematic depiction of Arc3A and Arc3B based on bioinformatic predictions. (**B**) Phylogeny of bacterial phyla with representative *arc* genes depicted and colored as in (**A**). Membrane-associated regions (grayed) and the number of predicted transmembrane segments are indicated. (**C**) Recovery of *P. aeruginosa* strains bearing the indicated mutations and containing a control chromosomal insertion (–) or insertion constructs expressing Arc3B proteins derived from assorted bacteria (*P. aer*, *P. aeruginosa*; *P. pro*, *P. protegens*; *B. cen*, *B. cenocepacia*; *P. syr*, *P. syringae*; *C. rod*, *Citrobacter rodentium*) following growth competition against *B. thai* wild-type. (**D**) Recovery of the indicated *P. protegens* strains following growth competition against *B. thai* wild-type or a strain lacking Tle3 activity. For interbacterial competition assays, the mean ± SD of biological duplicates and two or three associated technical replicates is shown: *p<0.05 using unpaired two-tailed Student's *t*-test.

The online version of this article includes the following figure supplement(s) for figure 3:

**Figure supplement 1.** EstA and Arc3B inactivation does not significantly impact *P. aeruginosa* fitness in pairwise growth competition experiments with *B. thai*.

plays an intrinsic role in defense, independent of its interaction Arc3B. We tested this by examining the ability of Arc3A-related proteins from other species to complement the defense defect of *P. aeruginosa* Δ*arc3A*. Despite high-sequence divergence that would likely preclude specific interactions with Arc3B or other *P. aeruginosa* factors, four of these proteins restored Tle3 defense to a statistically significant degree and two provided protection equivalent to that of *P. aeruginosa arc3A* expressed in the same manner (*Figure 3C*). We also examined one of these proteins in its native context. Inactivation of *arc3A* in *Pseudomonas protegens* conferred a strong interbacterial antagonism defense defect specific to *B. thai* Tle3 (*Figure 3D*). In total, these data strongly suggest that proteins in the Arc3A family possess the intrinsic capacity to protect bacteria against Tle3-like toxins, and potentially phospholipase toxins more broadly.

While the ability of heterologously expressed Arc3A homologs to contribute to *P. aeruginosa* defense demonstrates that these proteins can function independently, this finding does not rule out the possibility that in *P. aeruginosa* Arc3B enhances the defensive function of Arc3A. In two of our Tn-seq screens (*Figure 1A and C*, *Figure 2C*), we observed a significant difference in *arc3B* insertion frequency during competition with wild-type *B. thai* compared to the T6SS- or Tle3-inactivated mutants. However, in pairwise competition experiments with *B. thai,* we were unable to detect a significant fitness defect associated with *arc3B* inactivation in either the wild-type or the Δ*arc3A* background (*Figure 3—figure supplement 1*). These results suggest that Arc3B may play an auxiliary role in defense that is only detectable under the more stringent conditions of the Tn-seq screens, where *P. aeruginosa* strains compete with each other as well as *B. thai*.

Enzymes with PLA activity cleave glycerophospholipids at the *sn1* or *sn2* position, generating fatty acid and detergent-like lysophospholipid products (*Filkin et al., 2020*). For toxins like Tle3, which are delivered to target cells in exceedingly low quantities (*Fridman et al., 2020*; *Hernandez et al., 2020*), it is likely toxic products, rather than phospholipid depletion per se, that most immediately contribute to cell death. Based on the intrinsic capacity of Arc3A to grant protection against Tle3, a predicted PLA, we hypothesized that it directly mitigates damage caused by the toxic products of Tle3. To test this, we measured phospholipid content within extracts derived from competing *P. aeruginosa* and *B. thai* strains at a time point immediately prior to detectable changes in *P. aeruginosa* viability. Strikingly, relative to mixtures containing the wild-type organisms, those containing *P. aeruginosa* Δ*arc3A* possessed highly elevated levels of lysophosphatidylethanolamine (LPE) and monolysocardiolipin (MLCL) in a manner dependent on the activity of Tle3 in *B. thai* (*Figure 4A*). Lysophosphatidylglycerol, free fatty acid, and corresponding parent phospholipid levels were not measurably altered by Tle3 (*Figure 4—figure supplement 1*).

The accumulated lysophospholipids we observed could derive from *P. aeruginosa* or they could result from retaliatory activity against *B. thai*. To distinguish these possibilities, we subjected *P. aeruginosa* strains with radiolabeled phospholipids to Tle3 intoxication by unlabeled *B. thai* strains. Radiographic TLC analysis of the phospholipids generated in these strain competition mixtures confirmed that *P. aeruginosa* lacking Arc3B accumulates LPE and MLCL during Tle3 intoxication (*Figure 4B–D*). Tle3 belongs to a family of effectors for which there remains no definitively characterized member. We were unable to measure the activity of the enzyme in vitro; thus, to determine whether the accumulation of LPE and MLCL were a direct result of Tle3 activity, we quantified phospholipids in extracts derived from *B. thai* strains undergoing Tle3-based self-intoxication (*Figure 2—figure supplement 3*). Samples derived from mixtures composed of a *B. thai* strain with Tle3 and a derivative strain sensitized to Tle3 intoxication through cognate immunity gene inactivation accumulated LPE and MLCL, mirroring our interspecies competition findings (*Figure 4E*).

In *E. coli*, lysophospholipids are transported across the inner membrane by the dedicated transporter LplT, after which they are reacylated by Aas (*Zheng et al., 2017*). Both LplT and Aas have been shown to play a critical role in defending *E. coli* cells from phospholipase attack (*Lin et al., 2018*). In *P. aeruginosa,* LplT and the acylation domain of Aas are found in a single predicted polypeptide, encoded by PA3267 (*aas*). Interestingly, the *aas* gene was illuminated as a potential *P. aeruginosa* antagonism defense factor in two of the transposon mutant screens we performed and *aas* genes neighbor *arc3* genes in many bacteria (*Figure 2C*, *Figure 1—figure supplement 1B*, *Figure 4—figure supplement 2*, *Figure 1—source data 1*, *Figure 2—source data 1*). In pairwise interbacterial competition assays, we were unable to detect the impact of *aas* inactivation on *P. aeruginosa* fitness; however, we found that *P. aeruginosa* strains lacking both *aas* and *arc3A* exhibited a Tle3 defense

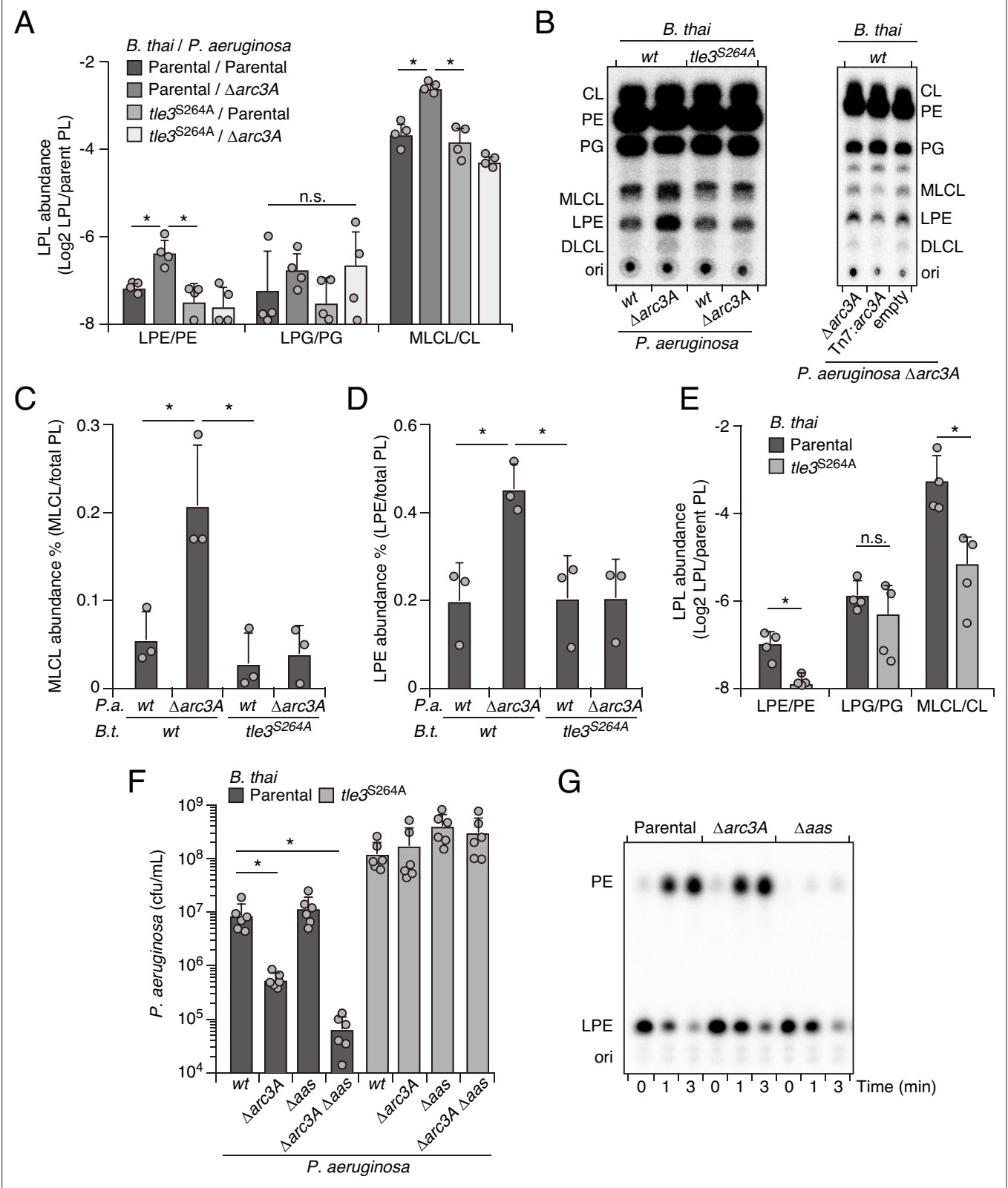

**Figure 4.** Arc3 prohibits Tle3-catalyzed lysophospholipid accumulation by a mechanism distinct from known pathways. (**A–D**) Mass spectrometric (**A**) and radiographic thin layer chromatography (TLC) (**B**) analysis of major phospholipid and lysophospholipid species within lipid extracts derived from the indicated mixtures of *P. aeruginosa* and *B. thai* strains. *P. aeruginosa* were grown in $^{32}PO_4^{2-}$ prior to incubation with *B. thai*. Strains were allowed to interact for 1 hr before phospholipids were harvested. Radiolabeled molecules of interest within biological triplicate experiments resolved by TLC were

*Figure 4 continued on next page*

*Figure 4 continued*

quantified by densitometry (**C, D**) (**E**) Mass spectrometric analysis of major phospholipid and lysophospholipid species within lipid extracts derived from mixtures containing *B. thai* lacking Tle3-specific immunity factors competing with the indicated *B. thai* strains. (**F**) Recovery of the indicated *P. aeruginosa* strains following growth competition against *B. thai* wild-type or a strain lacking Tle3 activity. (**G**) Radiographic TLC analysis of products extracted after incubation of *P. aeruginosa*-derived spheroplasts with purified radiolabeled lysophosphatidylethanolamine (LPE). For interbacterial competition assays, the mean ± SD of biological replicates and associated technical replicates is shown: *p<0.05 using unpaired two-tailed Student's *t*-test. TLC images shown are representative of at least two biological replicates. See also *Figure 4—source data 1–3*.

The online version of this article includes the following source data and figure supplement(s) for figure 4:

**Source data 1.** Original uncropped image and original uncropped image with relevant spots indicated for *Figure 4B*, left TLC.

**Source data 2.** Original uncropped image and original uncropped image with relevant spots indicated for *Figure 4B*, right TLC.

**Source data 3.** Original uncropped image and original uncropped image with relevant spots indicated for *Figure 4G*, right TLC.

**Figure supplement 1.** Tle3 intoxication does not impact free fatty acids nor intact parent phospholipids in *P. aeruginosa*.

**Figure supplement 2.** *Arc3B* genes are encoded adjacent to *aas* genes in diverse bacteria.

**Figure supplement 3.** Overexpression of *aas* fails to complement the competitive defect associated with Arc3A inactivation.

defect substantially greater than that associated with Arc3A inactivation alone (*Figure 4F*). Aas over-expression complements the Δ*aas*-dependent defect in this strain, but it does not compensate for the inactivation of Arc3A (*Figure 4—figure supplement 3*). Additionally, we found that Aas is required for PE regeneration from LPE, whereas Arc3A neither contributed to this process, nor could we detect changes in LPE stability upon Arc3A inactivation in the absence of antagonism (*Figure 4G*). These findings indicate that *arc3* encodes a previously undescribed, widespread and independent pathway that prohibits lysophospholipid accumulation by an interbacterial phospholipase toxin.

## Discussion

Evidence suggests that the existence of bacteria is characterized by an onslaught of challenges to their integrity (*Granato et al., 2019*; *Peterson et al., 2020*). This is borne out in part by recent work in the phage arena, which has highlighted a vast number of dedicated bacterial defense systems against these parasites (*Bernheim and Sorek, 2020*; *Hampton et al., 2020*). It is now appreciated that the amalgam of these systems constitutes a bacterial immune system, with both innate and adaptive arms that are functionally analogous, and even evolutionarily related to those of eukaryotic organisms (*Morehouse et al., 2020*). Our work shows that bacteria also possess a multitude of coordinately regulated pathways that provide defense against the threats posed by other bacteria. This further cements the analogy between the immune systems of bacterial and eukaryotic cells, the latter of which also rely on the coordinated production of factors specialized for countering threats of viral or bacterial origin.

We found that Arc3A is the single cellular factor of *P. aeruginosa* most critical for defense against a PLA toxin. Whether Arc3A, perhaps in concert with Arc3B, offers a wider breadth of defensive activity is not known. However, it is notable that phospholipase toxins are ancient and among the most prevalent bacterial toxins known. Even in the absence of direct antagonism by a PLA toxin, environmental lysophospholipids and those generated during normal cellular processes may represent ongoing threats to bacterial viability (*Flores-Díaz et al., 2016*). We observed low Arc3A expression in the absence of Gac/Rsm activation, arguing that its role in PARA serves an adaptive purpose.

The precise mechanism by which Arc3 proteins defend against intoxication by phospholipase toxins remains unknown. Although we found that Arc3 grants specific protection against Tle3, the relatively low levels of *P. aeruginosa* intoxication achieved by Tle1 do not permit us to rule out that it might also provide broader phospholipase defense. The number and density of transmembrane helices in Arc3A family proteins is unusual, and we postulate that these anomalous features of the protein are intimately connected to its mechanism of action. These segments may sequester lysophospholipids and facilitate their turnover or it is conceivable that they directly recruit and interfere with phospholipase enzymes.

We have found that several previously uncharacterized gene clusters under Gac/Rsm control contribute critically to PARA. Our bioinformatic analyses show that the *arc* clusters are widely conserved among Bacteria, yet the Gac/Rsm pathway is restricted to γ-Proteobacteria. If *arc* genes play similar

roles in these diverse bacteria, which our data suggest, their expression is also likely subject to induction by antagonism. It will be of interest in future studies to probe the generality of coordinated interbacterial defenses by identifying additional signaling systems responsive to antagonism. Collectively, our findings provide new insight into the diversity of mechanisms bacteria have evolved to defend against interbacterial attack and suggest that many more cryptic specialized defense pathways await discovery.

## Materials and methods

### Bacterial strains and culture conditions

A detailed list of all strains and plasmids used in this study can be found in Appendix 2—key resources table. Bacterial strains under investigation in this study were derived from *P. aeruginosa* PAO1, *B. thai* E264, and *P.protegens* pf-5 (*Paulsen et al., 2005*; *Stover et al., 2000*; *Yu et al., 2006*). All strains were grown in Luria-Bertani (LB) broth and incubated either at 37°C (*P. aeruginosa* and *B. thai*) or 30°C (*P. protegens*). For *Pseudomonas* strains, the media was supplemented with 25 µg/ml irgasan, 15 µg/ml gentamicin, 5% (*w/v*) sucrose, and 0.2% (*w/v*) arabinose as needed. *B. thai* was cultured with 15 µg/ml gentamicin and 200 µg/ml trimethoprim as necessary, and counterselection for allelic exchange was performed on M9 minimal medium agar plate containing 0.4% (*w/v*) glucose and 0.2% (*w/v*) *p*-chlorophenylalanine. *E. coli* strains used in this study included DH5α for plasmid maintenance, SM10 for conjugal transfer of plasmids into *P. aeruginosa*, *P. protegens*, and *B. thai,* and UE54 for radiolabeled phosphatidylethanolamine (PE) synthesis. *E. coli* strains were grown in LB supplemented with 15 µg/ml gentamicin, 150 µg/ml carbenicillin, and 200 µg/ml trimethoprim as needed.

### Plasmid construction

All primers used in plasmid construction and generation of mutant strains are listed in *Appendix 3—table 1*. In-frame chromosomal deletions and point mutations were created using the suicide vector pEXG2 for *P. aeruginosa* and *P. protegens*, and pJRC115 for *B. thai* (*Chandler et al., 2009*; *Rietsch et al., 2005*). For the generation of chromosomal mutation constructs, 750 bp regions flanking the mutation site were PCR amplified and inserted stitched together into the appropriate vector via Gibson assembly (*Gibson et al., 2009*). Site-specific chromosomal insertions in *P. aeruginosa* were generated using pUC18-Tn7t-pBAD-araE as previously described (*Hoang et al., 2000*). Our trimethoprim-resistant *B. thai* strain was generated using pUC18T-mini-Tn7-Tp-PS12-mCherry via the mini-Tn7 system (*LeRoux et al., 2012*).

### Generation of mutant strains

For *P. aeruginosa* and *P. protegens* strain generation, pEXG2 mutation constructs were transformed into *E. coli* SM10, and SM10 donors were subsequently mixed and incubated together with *Pseudomonas* recipients on a nitrocellulose membrane on top of an LB agar plate at a ratio of 10:1 donor–recipient. The cell mixtures were incubated for 6 hr at 37°C to allow for conjugation. These cell mixtures were then scraped up, resuspended into 200 µl LB, and plated on LB agar plates containing irgasan and gentamicin to select for cells containing the mutant construct inserted into the chromosome. *Pseudomonas* merodiploid strains were then grown overnight in nonselective LB media, followed by counterselection on LB no-salt agar plates supplemented with sucrose. Gentamicin-sensitive, sucrose-resistant colonies were screened for allelic replacement by colony PCR, and mutations were confirmed via Sanger sequencing of PCR products.

For *B. thai* strain generation, pJRC115 mutation constructs were transformed into *E. coli* SM10. Conjugation to enable plasmid transfer was performed as described above and plated on LB agar containing gentamicin and trimethoprim to select for *B. thai* with the chromosomally integrated plasmid. Merodiploids were grown overnight in LB media at 37°C, then plated on M9 minimal medium containing *p*-chlorophenylalanine for counterselection. Trimethoprim-sensitive, *p*-chlorophenylalanine-resistant colonies were screened for allelic replacement by colony PCR, and mutations were confirmed via Sanger sequencing of PCR products.

For expression of Arc1F, Arc2G, Arc3A, and Arc3A-homologs in *P. aeruginosa*, pUC18-Tn7t-pBAD-araE containing the gene of interest and the helper plasmid pTNS3 were co-transformed into *P.*

*aeruginosa* strains by electroporation (*Choi et al., 2008*). After 6 hr of outgrowth in LB at 37°C, transformants were plated on an LB agar plate with gentamicin to select for cells with mini-Tn7 integration.

## Bacterial competition assay

For each co-culture competition experiment, donor and recipient strains were first grown for 20 hr in LB medium. Cultures were spun for 1 min at 20,000 × *g* to pellet cells, the supernatant removed, and cells were washed once with fresh LB medium. Cell pellets were resuspended in LB and normalized to $OD_{600}$ = 20. Mixtures of donor and recipient strains were then established at 20:1 (*B. thai* vs. *P. aeruginosa, P. protegens,* or *B. thai*) or 10:1 (*P. aeruginosa* vs. *B. thai*) *v/v* ratios. The initial ratios of donor and recipient strains in these mixtures were measured by performing 10-fold serial dilutions and plating on appropriate selective media to evaluate by colony-forming unit (CFU) analysis. The co-culture competitions were initiated by spotting 5 µl of each mixture onto nitrocellulose filters placed on 3% (*w/v*) agar LB no-salt plates supplemented with L-arabinose for induction of gene expression as necessary. Competitions were incubated for 6 hr at 37°C between *B. thai* and *P. aeruginosa,* or 6 hr at 30°C for competition between *B. thai* and *P. protegens.* For *B. thai* self-intoxication, competitions were incubated at 37°C for 3 hr. Cells were harvested by scraping individual spots from excised sections of the nitrocellulose filter into LB medium. Suspensions were serially diluted and plated on selective media for CFU quantification.

## Immunoblotting analysis

To analyze the expression of Arc1F-VSV-G, Arc2G-VSV-G, and Arc3A-VSV-G, *P. aeruginosa* strains were grown in LB broth at 37°C to log phase. Cells were pelleted and resuspended in lysis buffer (20 mM Tris-HCl, pH 7.5, 300 mM NaCl, 10% [*v/v*] glycerol) and then mixed 1:1 with 2X SDS-PAGE sample loading buffer. Samples were then boiled at 100°C for 10 min and loaded at equal volumes to resolve using SDS-PAGE, then transferred to nitrocellulose membranes. Membranes were blocked in TBST (10 mM Tris-HCl pH 7.5, 150 mM NaCl, and 0.1% [*w/v*] Tween-20) with 5% (*w/v*) non-fat milk for 30 min at room temperature, followed by incubation with anti-VSV-G or with anti-ribosome polymerase β subunit primary antibodies diluted in TBST for 1 hr at room temperature on an orbital shaker. Blots were then washed with TBST, followed by incubation with secondary antibody (goat anti-rabbit or anti-mouse HRP conjugated) diluted in TBST for 30 min at room temperature. Finally, blots were washed with TBST again and developed using Radiance HRP substrate and visualized using iBright imager.

## Lipidomic analysis

To analyze lipid content after intra- and inter-species intoxication, overnight cultures of donor and recipient cells were harvested and mixed at 1:1 (*v/v*) ratio in LB medium. These mixtures were spotted onto a 3% (*w/v*) agar LB no-salt plate and incubated for 1 hr at 37°C to allow for intoxication. Cells were then collected, and the lipids were extracted using the Bligh–Dyer method (*Bligh and Dyer, 1959*). Briefly, bacterial pellets were resuspended in 0.5 ml of solution containing 0.5 M NaCl in 0.5 N HCl, followed by adding 1.5 ml of chloroform/methanol mixture (1:2, *v/v*). Suspensions were vortexed at room temperature for 15 min, 1.5 ml of 0.5 M NaCl in 0.5 N HCl solution was added to each sample, and vortexed for an additional 5 min. Lipid and aqueous layers were separated by centrifugation at 2000 × *g* for 5 min. The lower lipid phase was collected, and these dried lipid samples were analyzed for PE, PG, CL, LPE, LPG, and MLCL content by the Kansas State Lipidomics Research Center.

Lipid samples were analyzed by electrospray ionization triple quadrupole mass spectrometry in direct infusion mode, and data were processed as described by with slight modifications (*Shiva et al., 2013*), which include use of a Waters Xevo TQS mass spectrometer (Milford, MA). Internal standards used to normalize data are shown in *Appendix 1—table 2*. For PE and LPE, SPLASH standards were used for normalization (Avanti Polar Lipids, Alabaster, AL). Mass spectrometry global parameters, scan modes, and data processing parameters are shown in *Appendix 1—table 2*.

## Proteomic analysis

*P. aeruginosa* strains containing VSV-G-tagged proteins were grown in 50 ml to mid-log phase, centrifuged at 2500 × *g* for 15 min, and the pellets harvested by resuspension in lysis buffer (20 mM Tris-HCl pH 7.5, 300 mM NaCl, 5% [*w/v*] glycerol, 0.5% [*v/v*] Triton X-100). Cell lysates were prepared by

sonication, and tagged proteins were enriched by incubating cell lysates with 30 µl of anti-VSV-G agarose beads at 4°C for 4 hr with constant rotation. Agarose beads were then pelleted by centrifugation at 100 × $g$ for 2 min and washed three times with 20 mM ammonium bicarbonate. VSV-G agarose beads and bound proteins were then treated with 10 µl of 10 ng/µl (100 ng total per sample) sequencing grade trypsin for 16 hr at 37°C. After digestion, 40 µl of 20 mM ammonium bicarbonate was added to the agarose beads and peptide mixture and lightly mixed. Beads were centrifuged at 300 × $g$ for 3 min, and the supernatant collected as the peptide fraction. This peptide mixture was reduced with 5 mM Tris(2-carboxyethyl) phosphine hydrochloride for 1 hr, followed by alkylation using 14 mM iodoacetamide for 30 min in the dark at room temperature. Alkylation reactions were quenched using 5 mM 1,4-dithiothreitol. Samples were then diluted with 100% acetonitrile (ACN) and 10% ($w/v$) trifluoroacetic acid (TFA) to a final concentration of 5% ACN ($v/v$) and 0.5% TFA ($w/v$) and applied to MacroSpin C18 columns (30 µg capacity) that had been charged with 100% ACN and ddH$_2$O. Bound peptides were then washed twice in 5% ($v/v$) ACN and 0.5% ($w/v$) TFA, before elution with 70% ($v/v$) ACN and 0.1% ($v/v$) formic acid (FA).

Peptides were analyzed by LC-MS/MS using a Dionex UltiMate 3000 Rapid Separation nanoLC and a Q Exactive HF Hybrid Quadrupole-Orbitrap Mass Spectrometer (Thermo Fisher Scientific Inc, San Jose, CA). Approximately 1 µg of peptide samples was loaded onto the trap column, which was 150 µm × 3 cm in-house packed with 3 µm C18 beads. The analytical column was a 75 µm × 10.5 cm PicoChip column packed with 3 µm C18 beads (New Objective, Inc, Woburn, MA). The flow rate was kept at 300 nl/min. Solvent A was 0.1% FA in water and Solvent B was 0.1% FA in ACN. The peptide was separated on a 120 min analytical gradient from 5% ACN/0.1% FA to 40% ACN/0.1% FA. The mass spectrometer was operated in data-dependent mode. The source voltage was 2.10 kV, and the capillary temperature was 320°C. MS 1 scans were acquired from 300 to 2000 $m/z$ at 60,000 resolving power and automatic gain control (AGC) set to 3 × 10$^6$. The top 15 most abundant precursor ions in each MS 1 scan were selected for fragmentation. Precursors were selected with an isolation width of 2 Da and fragmented by higher-energy collisional dissociation (HCD) at 30% normalized collision energy in the HCD cell. Previously selected ions were dynamically excluded from re-selection for 20 s. The MS 2 AGC was set to 1 × 10$^5$.

Proteins were identified from the tandem mass spectra extracted by Xcalibur version 4.0. MS/MS spectra were searched against the Uniprot *P. aeruginosa* PAO1 strain database using Mascot search engine (Matrix Science, London, UK; version 2.5.1). The MS 1 precursor mass tolerance was set to 10 ppm, and the MS 2 tolerance was set to 0.05 Da. A 1% false discovery rate cutoff was applied at the peptide level. Only proteins with a minimum of two unique peptides above the cutoff were considered for further study. The search result was visualized by Scaffold (version 4.8.3, Proteome Software, Inc, Portland, OR).

## Transposon mutant library construction

A *P. aeruginosa* PAO1 transposon mutant library of ~80,000 unique Himar1 insertions was prepared using established protocols (*Lee et al., 2017*). Briefly, *P. aeruginosa* were mutagenized by delivery of the transposon- and transposase-bearing suicide vector pBT20 from *E. coli* SM10 $\lambda$ pir (*Kulasekara et al., 2005*). Insertion mutants were selected on LB agar containing irgasan and gentamycin and pooled. The library was harvested and transposon insertion sites of the complete pool were defined by transposon insertion sequencing as described below.

## Transposon mutant library screen in bacterial competition

The *P. aeruginosa* transposon mutant library was grown at 37°C to mid-log phase. Cell suspensions were then normalized to OD$_{600}$ = 1 and incubated with 100 OD$_{600}$ of *B. thai* donor strains (wild-type, Δ*icmF*, *tle3*$^{S264A}$, or Δ*colA*). Competitions were performed on 3% ($w/v$) agar LB no-salt plates at 37°C for 7 hr. Cells were washed once with PBS, diluted, and plated on LB agar plates supplemented with irgasan to select for viable *P. aeruginosa* cells. After overnight incubation, cells were harvested, and genomic DNA was directly extracted from pellets using QIAGEN Blood and Tissue Midi gDNA prep kit. Transposon insertion sequencing libraries were generated from 3 µg gDNA per sample using the C-tailing method as described (*Gallagher, 2019*) with the transposon-specific primers listed in *Appendix 3—table 1*. For PCR round 1, PCR_1A and PCR_1B were used. For PCR round 2, PCR_2A,

PCR_2B, and PCR_2C were used. For sequencing, Seq_primer was used. Libraries were pooled and sequenced in multiplex using an Illumina MiSeq with a 5% PhiX spike-in.

## Tn-seq data analysis

Seqmagick was used to trim the first six bases from each read (https://github.com/fhcrc/seqmagick; *Matsen Group, 2020*). Reads were then aligned, and counts were enumerated using TRANSIT TPP (https://transit.readthedocs.io/en/latest/tpp.html). An annotation GFF was created from the original *P. aeruginosa* PAO1 GFF file (GCF_000006765.1) and was translated to TRANSIT portable format using the TRANSIT 'convert gff_to_prot_table' command (https://transit.readthedocs.io/en/latest/transit_running.html#prot-tables-annotations). Finally, the TRANSIT 'export combined_wig' command was used to combine and annotate the counts. *Appendix 1—table 3* provides summary data of the sequencing runs and their processing by TRANSIT.

## Bioinformatic analysis of Arc3 gene distribution

The phylogenetic profiler tool in IMG (*Chen et al., 2021*) was used to determine the distribution of Arc3A homologs (identified as proteins classified under the Pfam 10101) across bacterial phyla. The number and location of predicted transmembrane domains present in representative Arc3A homologs from each phylum were calculated using CCTOP and TMpred (https://embnet.vital-it.ch/software/TMPRED_form.html) (*Dobson et al., 2015*). Homologs of Arc3B encoded adjacent to Arc3A-like genes were identified on the basis of their classification in the Pfam 13,163.

## Bacterial stress assays

*P. aeruginosa* cultures were assayed for the ability to survive oxidative stress (50 mM $H_2O_2$), heat stress (55°C), osmotic stress (3 M NaCl), and detergent stress (0.5% [*w/v*] SDS and 0.25 mM EDTA) in LB media as previously described with modifications (*Jørgensen et al., 1999*). After incubation with the oxidative, heat, and osmotic stress conditions, cultures were serially diluted 10-fold in LB medium and plated on LB agar for subsequent analysis of remaining viable cells by CFU determination. For oxidative stress, stationary-phase cultures were diluted to $OD_{600}$ = 0.1 and incubated in LB media containing 50 mM $H_2O_2$ for 30 min at 37°C with continuous shaking. For heat stress, stationary-phase cultures were diluted in pre-warmed LB media to $OD_{600}$ = 0.1 and incubated at 55°C for 30 min with continuous shaking. For osmotic stress, overnight cultures were diluted in LB media containing 3 M NaCl to $OD_{600}$ = 0.1. Cultures were incubated at 37°C for 20 hr with shaking. For detergent stress, cultures were serially diluted in LB media and directly plated on LB agar plates containing 0.5% (*w/v*) SDS and 0.25 mM EDTA for CFU quantification.

## Phospholipid analysis

To prepare [$^{32}$P]-labeled bacteria, *P. aeruginosa* strains were grown in LB media supplemented with 5 µCi/ml [$^{32}$P]-orthophosphoric acid. Tle3 intoxication was performed by mixing 200 $OD_{600}$ of *B. thai* and [$^{32}$P]-*P. aeruginosa* at a 10:1 ratio (*v/v*) and incubating this mixture at 37°C for 1 hr on 3% (*w/v*) agar LB no-salt plates containing L-arabinose as necessary. After 1 hr of intoxication, cells were collected, and the lipids were extracted using the Bligh–Dyer method as described above. Purified lipid samples were loaded onto a silica gel thin-layer plate and developed with chloroform/methanol/acetic acid/ddH$_2$O (85:15:10:3.5, *v/v/v/v*) solvent system (*Lopalco et al., 2017*). The air-dried plate was exposed to a storage phosphor screen. Individual phospholipid was visualized and quantified using phosphorimaging to calculate phospholipid content expressed as mol% of the total phospholipid pool.

## PE regeneration assay

To generate [$^{32}$P]-labeled PE, *E. coli* UE54 strain was grown in LB media supplemented with 5 µCi/ml [$^{32}$P]-orthophosphate overnight at 37°C (*Harvat et al., 2005*). Cells were collected and the lipids were extracted using the Bligh–Dyer method as described above. The extracted lipids were dried and resuspended in buffer containing 100 mM HEPES-NaOH, pH = 7.5, 100 mM KCl, 10 mM CaCl$_2$, and 1% (*w/v*) *n*-Dodecyl-β-D-maltoside. 10 units of pancreas phospholipase A$_2$ were added to digest PE to LPE at 37°C overnight with shaking. After incubation, lipids were extracted and loaded onto a silica gel thin-layer plate and developed with chloroform/methanol/acetic acid/ddH$_2$O (85:15:10:3.5, *v/v/v/v*) solvent system. The dried plate was exposed to an x-ray film for 2 hr. The phospholipid bands

were visualized by developing the film, and bands corresponding to [$^{32}$P]-LPE on the TLC plate were scraped, extracted, and resuspended in 100% ethanol.

*P. aeruginosa* spheroplasts were generated by resuspending log-phase culture in 25 mM Tris-HCl, pH 8, 450 mM sucrose, and 1.4 mM EDTA. After addition of 20 µg/ml lysozyme, cells were incubated on ice for 30 min. Intact spheroplasts were collected by centrifugation (3000 × *g* for 5 min) at 4°C and gently resuspended in 25 mM Tris-HCl, pH 8, and 450 mM sucrose. To examine PE regeneration, [$^{32}$P]-LPE was added into spheroplast solutions and incubated at 37°C. The reactions were terminated at the indicated time by adding chloroform-methanol mixture (1:2, *v/v*). The lipids were extracted, separated by TLC, and analyzed using phosphorimaging as described above (*Lin et al., 2019*).

## Statistical analyses

GraphPad Prism 7 software (San Diego, CA) was used for statistical analysis. We used an unpaired two-tailed Student's *t*-test for pairwise comparisons. In all cases, $p < 0.05$ was considered statistically significant.

## Acknowledgements

We thank Simon Dove, Josh Woodward, Lei Zheng, and Mougous laboratory members for helpful discussions, Mikhail Bogdanov and Lei Zheng for sharing reagents, and Colin Manoil and Jason Smith for sharing equipment. This work was supported by the grants from NIH (AI080609 to JDM, DK089507 to SJS, and R01AI136979 to LX) and the Cystic Fibrosis Foundation (SINGH19R0). Equipment utilized was supported by the Office of the Director, National Institutes of Health, under award number S10OD026741. JDM is an HHMI Investigator.

## Additional information

### Funding

| Funder | Grant reference number | Author |
| --- | --- | --- |
| National Institutes of Health | AI080609 | Joseph D Mougous |
| National Institutes of Health | DK089507 | Stephen J Salipante |
| National Institutes of Health | R01AI136979 | Libin Xu |
| Cystic Fibrosis Foundation | SINGH19R0 | Stephen J Salipante |
| National Institutes of Health | S10OD026741 | Stephen J Salipante |
| Howard Hughes Medical Institute | | Joseph D Mougous |

The funders had no role in study design, data collection and interpretation, or the decision to submit the work for publication.

### Author contributions

See-Yeun Ting, Conceptualization, Data curation, Writing - original draft, Writing - review and editing; Kaitlyn D LaCourse, Hannah E Ledvina, Conceptualization, Data curation; Rutan Zhang, Matthew C Radey, Hemantha D Kulasekara, Rahul Somavanshi, Savannah K Bertolli, Larry A Gallagher, Jennifer Kim, Kelsi M Penewit, Stephen J Salipante, Libin Xu, Data curation; S Brook Peterson, Conceptualization, Writing - original draft, Writing - review and editing; Joseph D Mougous, Conceptualization, Funding acquisition, Supervision, Writing - original draft, Writing - review and editing

### Author ORCIDs

Libin Xu (iD) http://orcid.org/0000-0003-1021-5200
S Brook Peterson (iD) http://orcid.org/0000-0003-2648-0965

Joseph D Mougous 🔗 http://orcid.org/0000-0002-5417-4861

**Decision letter and Author response**
Decision letter https://doi.org/10.7554/eLife.74658.sa1
Author response https://doi.org/10.7554/eLife.74658.sa2

## Additional files

### Supplementary files
• Transparent reporting form

### Data availability
Sequence data associated with this study is available from the Sequence Read Archive at BioProject PRJNA754428.

The following dataset was generated:

| Author(s) | Year | Dataset title | Dataset URL | Database and Identifier |
|---|---|---|---|---|
| Ting SY, Lacourse KD, Ledvina HE, Zhang R, Radey MC, Kulasekara HD, Somavanshi R, Bertolli SK, Gallagher LA, Kim J, Penewit KM, Salipante SJ, Xu L, Peterson SB, Mougous JD | 2021 | Coordinately regulated interbacterial antagonism defense pathways constitute a bacterial innate immune system | http://www.ncbi.nlm.nih.gov/bioproject/754428 | NCBI BioProject, PRJNA754428 |

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

# Appendix 1

**Appendix 1—table 1.** Mass spectrometric analysis of *P. aeruginosa* Arc3A and Arc3B immunoprecipitation samples.

| Locus ID* | Peptide counts of sample (%)[†] | Fold change (VSV-G/Ctrl [‡]) |
|---|---|---|
| Arc3B-VSV-G sample | | |
| **PA5113 (Arc3B)** | **5.86** | **> 200** |
| PA4385 | 1.62 | 2.29 |
| PA5016 | 1.45 | 0.82 |
| PA0090 | 1.17 | 0.72 |
| PA3729 | 1.12 | 1.30 |
| PA4761 | 1.12 | 0.71 |
| **PA5114 (Arc3A)** | **1.12** | **N.D. [§]** |
| PA5239 | 1.06 | 1.10 |
| PA3861 | 1.00 | 1.10 |
| PA4269 | 1.00 | 1.10 |
| PA1092 | 0.95 | 0.78 |
| PA0141 | 0.89 | 0.71 |
| PA0963 | 0.84 | 1.27 |
| PA2976 | 0.84 | 0.83 |
| PA4260 | 0.84 | 0.69 |
| PA4265 | 0.84 | 0.61 |
| PA4270 | 0.84 | 1.04 |
| PA1803 | 0.78 | 0.74 |
| PA2151 | 0.78 | 0.77 |
| PA3950 | 0.78 | 1.19 |
| PA3001 | 0.73 | 1.10 |
| PA3656 | 0.73 | 0.80 |
| PA5173 | 0.73 | 0.76 |
| PA5554 | 0.73 | 1.20 |
| PA5556 | 0.73 | 2.05 |
| Arc3A-VSV-G sample | | |
| PA5554 | 2.92 | 4.81 |
| **PA5113 (Arc3B)** | **2.22** | **> 200** |
| **PA5114 (Arc3A)** | **1.44** | **N.D.** |
| PA4751 | 1.30 | 12.87 |
| PA5556 | 1.26 | 3.55 |
| PA3729 | 1.10 | 1.28 |
| PA5016 | 1.10 | 0.62 |
| PA4429 | 0.99 | N.D. |
| PA4942 | 0.96 | 6.36 |
| PA2493 | 0.92 | 2.60 |
| PA3160 | 0.90 | N.D. |
| PA4246 | 0.87 | 0.79 |

*Appendix 1—table 1 Continued on next page*

*Appendix 1—table 1 Continued*

| Locus ID* | Peptide counts of sample (%)[†] | Fold change (VSV-G/Ctrl [‡]) |
|---|---|---|
| PA0090 | 0.85 | 0.53 |
| PA1552 | 0.83 | N.D. |
| PA0077 | 0.81 | 5.33 |
| PA2494 | 0.81 | ND |
| PA2976 | 0.72 | 0.71 |
| PA0659 | 0.70 | N.D. |
| PA4761 | 0.70 | 0.44 |
| PA4941 | 0.65 | 2.57 |
| PA4265 | 0.63 | 0.46 |
| PA3656 | 0.61 | 0.67 |
| PA3794 | 0.61 | N.D. |
| PA2151 | 0.56 | 0.55 |
| PA3821 | 0.56 | N.D. |

* Proteins containing at least two peptides identified in a given immunoprecipitation sample are included. Arc3A and Arc3B are bolded.

[†] Value corresponds to the abundance (peptide counts) of the protein within the total immunoprecipitation sample. Only the 25 most abundant proteins in each sample are shown.

[‡] Immunoprecipitation from a *P. aeruginosa* strain lacking a VSV-G-tagged protein served as the control sample (Ctrl).

[§] The protein was not detected in the control sample.

**Appendix 1—table 2.** Internal standards and parameters used in lipidomic analysis.

| Internal standards used | | |
|---|---|---|
| **Compound formula** | **Compound Name** | **Alternative name** |
| C20H41O9P | LysoPG(14:0) | |
| C24H49O9P | LysoPG(18:0) | |
| C34H67O10P | PG(14:0/14:0) | |
| C46H91O10P | PG(20:0/20:0), i.e. diphytanoyl PG | |
| C19H40O7PN | LysoPE(14:0) | |
| C23H48O7PN | LysoPE(18:0) | |
| C29H58O8PN | PE(12:0/12:0) | |
| C45H90O8PN | PE(20:0/20:0), i.e. diphytanoyl PE | |
| C66H120O17P2 | CL(57:4) | 14:1(3)–15:1 CA |
| C70H134O17P2 | CL(61:1) | 15:0(3)–16:1 CA |
| C89H166O17P2 | CL(80:4) | 22:1(3)–14:1 CA |

| Mass spectrometry global parameters on Waters Xevo TQS mass spectrometer | |
|---|---|
| Parameter | Value |
| Cycle time | Automatic |
| Source temperature (°C) | 150 |
| Desolvation temperature (°C) | 250 |
| Cone gas flow (L/h) | 150 |
| Desolvation gas fow (L/h) | 650 |
| Collision gas flow (mL/min) | 0.14 |
| Nebuliser gas (Bar) | 7 |
| LM 1 Resolution | 2.8 |
| HM 1 Resolution | 14.8 |

### Mass spectral acquisition and data processing parameters

#### Acquisition parameters

| Compound | Function | m/z range | Start/end time (min) | Scan duration (s) | Ionization mode | Cone voltage (V) | Collision energy (V) | Total scan time (min) |
|---|---|---|---|---|---|---|---|---|
| PG | Neutral Loss of 189.04 | 680–880 | 1.6–5 | 2 | ES+ | 30 | 8 | 3.4 |
| LPE, PE | Neutral Loss of 141.02 | 422–922 | 1.6–5 | 5 | ES+ | 40 | 20 | 3.4 |
| LPG | Parents of 152.9 | 420–570 | 5.2–10 | 0.75 | ES- | 30 | 38 | 4.8 |
| CL, MLCL | Parents of 153 | 865–1,572 | 1.6–30 | 4.7 | ES- | 40 | 75 | 28.4 |

#### Data processing parameters

| Subtraction | Smooth mean | Centroid mid peak width at half height (all top 50%) |
|---|---|---|
| 1, 40, 0.01 | 2 × 0.4 | 2 |
| 1, 40, 0.01 | 2 × 0.4 | 2 |
| 1, 40, 0.01 | 1 × 0.6 | 5 |
| 1, 40, 0.01 | 2 × 0.6 | 5 |

**Appendix 1—table 3.** Transposon sequencing data summary.

| Growth experiment | Mutant pool pre-competition | Pool with WT *B. thai* - rep 1 | Pool with δt6s *B. thai* - rep 1 |
|---|---|---|---|
| fastq file name | tn_mutant_pool.fastq | pool_v_wt_rep1.fastq | pool_v_deltaT6_rep1.fastq |
| Total reads | 515,434 | 5,788,728 | 6,452,899 |
| Trimmed (valid Tn prefix) | 515,293 | 5,786,427 | 6,450,594 |
| Mapped | 350,923 | 4,666,597 | 5,283,645 |
| Mapped to TA sites | 349,915 | 4,634,942 | 5,241,672 |
| TA_sites | 94,404 | 94,404 | 94,404 |
| TAs_hit | 50,348 | 57,644 | 60,331 |
| Max reads per TA site | 324 | 9,519 | 5,886 |
| Mean reads per hit TA site | 6.9 | 80.4 | 86.9 |

| Pool with WT *B.thai* - rep 2 | Pool with ΔT6S *B.thai* - rep 2 | Pool with *tle3*S264A *B. thai* | Pool with Δ*colA B. thai* |
|---|---|---|---|
| pool_v_wt_rep2.fastq | pool_v_deltaT6_rep2.fastq | pool_v_tle3.fastq | pool_v_colA.fastq |
| 411,515 | 720,133 | 3,226,978 | 3,130,545 |
| 411,401 | 719,895 | 3,225,816 | 3,129,574 |
| 266,466 | 481,466 | 3,024,047 | 2,938,318 |
| 265,661 | 479,769 | 2,997,466 | 2,922,995 |
| 94,404 | 94,404 | 94,404 | 94,404 |

*Appendix 1—table 3 Continued on next page*

*Appendix 1—table 3 Continued*

| Growth experiment | Mutant pool pre-competition | Pool with WT *B. thai* - rep 1 | Pool with δt6s *B. thai* - rep 1 |
|---|---|---|---|
| 46,500 | 53,148 | 59,145 | 58,101 |
| 669 | 447 | 3,484 | 3,568 |
| 5.7 | 9.0 | 50.7 | 50.3 |

# Appendix 2

**Appendix 2—key resources table**

| Reagent type (species) or resource | Designation | Source or reference | Identifiers | Additional information |
|---|---|---|---|---|
| Strain, strain background (*Pseudomonas aeruginosa*) | PAO1 | *Stover et al., 2000* | | |
| Strain, strain background (*P. aeruginosa*) | PAO1 Δ*pa2541* | This study | | See Materials and methods |
| Strain, strain background (*P. aeruginosa*) | PAO1 Δ*pa2541Δpa2536* | This study | | See Materials and methods |
| Strain, strain background (*P. aeruginosa*) | PAO1 Δ*pa4323* | This study | | See Materials and methods |
| Strain, strain background (*P. aeruginosa*) | PAO1 Δ*pa4323Δpa4320* | This study | | See Materials and methods |
| Strain, strain background (*P. aeruginosa*) | PAO1 Δ*pa5114* | This study | | See Materials and methods |
| Strain, strain background (*P. aeruginosa*) | PAO1 Δ*pa5114Δpa5113* | This study | | See Materials and methods |
| Strain, strain background (*P. aeruginosa*) | PAO1 Δ*pa2541, Tn7::AraE::pa2541* | This study | | See Materials and methods |
| Strain, strain background (*P. aeruginosa*) | PAO1 Δ*pa4323, Tn7::AraE::pa4323* | This study | | See Materials and methods |
| Strain, strain background (*P. aeruginosa*) | PAO1 Δ*pa5114, Tn7::AraE::pa5114* | This study | | See Materials and methods |
| Strain, strain background (*P. aeruginosa*) | PAO1 Δ*pa2541Δpa4323Δpa5114* | This study | | See Materials and methods |
| Strain, strain background (*P. aeruginosa*) | PAO1 Δ*gacS* (Δ*pa0928*) | *LeRoux et al., 2015a* | | |
| Strain, strain background (*P. aeruginosa*) | PAO1 Δ*icmF1* (Δ*pa0077*) | *Silverman et al., 2011* | | |
| Strain, strain background (*P. aeruginosa*) | PAO1 Δ*pa5112* | This study | | See Materials and methods |
| Strain, strain background (*P. aeruginosa*) | PAO1 Δ*pa5114, Tn7::AraE::empty* | This study | | See Materials and methods |
| Strain, strain background (*P. aeruginosa*) | PAO1 Δ*pa5114, Tn7::AraE:: pfl_rs01995* | This study | | See Materials and methods |

*Appendix 2 Continued on next page*

*Appendix 2 Continued*

| Reagent type (species) or resource | Designation | Source or reference | Identifiers | Additional information |
|---|---|---|---|---|
| Strain, strain background (*P. aeruginosa*) | PAO1 Δpa5114, Tn7::AraE:: i35_rs27920 | This study | | See Materials and methods |
| Strain, strain background (*P. aeruginosa*) | PAO1 Δpa5114, Tn7::AraE::pspto_rs26695 | This study | | See Materials and methods |
| Strain, strain background (*P. aeruginosa*) | PAO1 Δpa5114, Tn7::AraE::ta05_rs00690 | This study | | See Materials and methods |
| Strain, strain background (*P. aeruginosa*) | PAO1 Δpa3267 | This study | | See Materials and methods |
| Strain, strain background (*P. aeruginosa*) | PAO1 Δpa5114Δpa3267 | This study | | See Materials and methods |
| Strain, strain background (*P. aeruginosa*) | PAO1 ΔretS (Δpa4856) | *Mougous et al., 2006* | | |
| Strain, strain background (*P. aeruginosa*) | PAO1 ΔretS Δpa3267 | This study | | See Materials and methods |
| Strain, strain background (*P. aeruginosa*) | PAO1 ΔretS Δpa5114 | This study | | See Materials and methods |
| Strain, strain background (*P. aeruginosa*) | PAO1 ΔrpoS (Δpa3622) | *Jørgensen et al., 1999* | | |
| Strain, strain background (*P. aeruginosa*) | PAO1 ΔvacJ (Δpa2800) | *Munguia et al., 2017* | | |
| Strain, strain background (*P. aeruginosa*) | PAO1 pa2541_vsv-g | This study | | See Materials and methods |
| Strain, strain background (*P. aeruginosa*) | PAO1 ΔretS, pa2541_vsv-g | This study | | See Materials and methods |
| Strain, strain background (*P. aeruginosa*) | PAO1 ΔgacS, pa2541_vsv-g | This study | | See Materials and methods |
| Strain, strain background (*P. aeruginosa*) | PAO1 pa4323_vsv-g | This study | | See Materials and methods |
| Strain, strain background (*P. aeruginosa*) | PAO1 ΔretS, pa4323_vsv-g | This study | | See Materials and methods |
| Strain, strain background (*P. aeruginosa*) | PAO1 ΔgacS, pa4323_vsv-g | This study | | See Materials and methods |
| Strain, strain background (*P. aeruginosa*) | PAO1 pa5114_vsv-g | This study | | See Materials and methods |

*Appendix 2 Continued on next page*

*Appendix 2 Continued*

| Reagent type (species) or resource | Designation | Source or reference | Identifiers | Additional information |
|---|---|---|---|---|
| Strain, strain background (*P. aeruginosa*) | PAO1 Δ*retS*, *pa5114_vsv-g* | This study | | See Materials and methods |
| Strain, strain background (*P. aeruginosa*) | PAO1 Δ*gacS*, *pa5114_vsv-g* | This study | | See Materials and methods |
| Strain, strain background (*P. aeruginosa*) | PAO1 Δ*retS*, *pa5113_vsv-g* | This study | | See Materials and methods |
| Strain, strain background (*Burkholderia thailandensis*) | E264 (ATCC 700388) | *Yu et al., 2006* | | |
| Strain, strain background (*B. thailandensis*) | E264 Δ*icmF1* (Δ*BTH_I2954*) | *LeRoux et al., 2015a* | | |
| Strain, strain background (*B. thailandensis*) | E264 *tle3*$^{S264A}$ (*BTH_II0090*$^{S264A}$ + *BTH_I3226*$^{S264A}$) | This study | | See Materials and methods |
| Strain, strain background (*B. thailandensis*) | E264 *tle1*$^{S267A}$ (*BTH_I2698*$^{S267A}$ + *BTH_I2701*$^{S267A}$) | This study | | See Materials and methods |
| Strain, strain background (*B. thailandensis*) | E264 Δ*colA* (Δ*BTH_I2691*) | This study | | See Materials and methods |
| Strain, strain background (*B. thailandensis*) | E264 Δ*tae2* (Δ*BTH_I0068*) | *Russell et al., 2012* | | |
| Strain, strain background (*B. thailandensis*) | E264 *Tn7::Tp-PS12-mCherry* | *LeRoux et al., 2012* | | |
| Strain, strain background (*B. thailandensis*) | E264 *tle3* sensitized strain (Δ*BTH_I3225-8*, Δ*BTH_II0089-94*) | This study | | See Materials and methods |
| Strain, strain background (*B. thailandensis*) | *E264 Tn7::Tp-PS12-mCherry tle3 sensitized strain (ΔBTH_I3225-8, ΔBTH_II0089-94)* | This study | | See Materials and methods |
| Strain, strain background (*Pseudomonas protegens*) | pf-5 (ATCC BAA-477) | *Paulsen et al., 2005* | | |
| Strain, strain background (*P. protegens*) | *Pseudomonas protegens pf-5 Δpfl_rs01995* | This study | | See Materials and methods |
| Strain, strain background (*Escherichia coli*) | DH5α | Thermo Fisher Scientific Cat# 18258012 | | |
| Strain, strain background (*E. coli*) | SM10 | Biomedal Lifescience Cat# BS-3303 | | |

*Appendix 2 Continued on next page*

*Appendix 2 Continued*

| Reagent type (species) or resource | Designation | Source or reference | Identifiers | Additional information |
|---|---|---|---|---|
| Strain, strain background (*E. coli*) | UE54 MG1655 *lpp-2Δara714 rcsF::mini-Tn10 cam pgsA::FRT-kan-FRT* | **Harvat et al., 2005** | | |
| Recombinant DNA reagent | pEXG2-Δ*pa2541* (plasmid) | This study | | Deletion construct, see Materials and methods |
| Recombinant DNA reagent | pEXG2-Δ*pa2536* (plasmid) | This study | | Deletion construct, see Materials and methods |
| Recombinant DNA reagent | pEXG2-Δ*pa4323* (plasmid) | This study | | Deletion construct, see Materials and methods |
| Recombinant DNA reagent | pEXG2-Δ*pa4320* (plasmid) | This study | | Deletion construct, see Materials and methods |
| Recombinant DNA reagent | pEXG2-Δ*pa5114* (plasmid) | This study | | Deletion construct, see Materials and methods |
| Recombinant DNA reagent | pEXG2-Δ*pa5114/pa5113* (plasmid) | This study | | Deletion construct, see Materials and methods |
| Recombinant DNA reagent | pEXG2-Δ*pa3267* (plasmid) | This study | | Deletion construct, see Materials and methods |
| Recombinant DNA reagent | pEXG2-Δ*pa5113* (plasmid) | This study | | Deletion construct, see Materials and methods |
| Recombinant DNA reagent | pEXG2-Δ*pa5112* (plasmid) | This study | | Deletion construct, see Materials and methods |
| Recombinant DNA reagent | pEXG2-Δ*pfl_rs01995* (plasmid) | This study | | Deletion construct, see Materials and methods |
| Recombinant DNA reagent | pEXG2-*pa2541_vsv-g* (plasmid) | This study | | Construct to introduce VsvG tag, see Materials and methods |
| Recombinant DNA reagent | pEXG2-*pa4323_vsv-g* (plasmid) | This study | | Construct to introduce VsvG tag, see Materials and methods |
| Recombinant DNA reagent | pEXG2-*pa5114_vsv-g* (plasmid) | This study | | Construct to introduce VsvG tag, see Materials and methods |
| Recombinant DNA reagent | pEXG2-*pa5113_vsv-g* (plasmid) | This study | | Construct to introduce VsvG tag, see Materials and methods |
| Recombinant DNA reagent | pUC18-Tn7t-pBAD-araE (plasmid) | **Hoang et al., 2000** | | Arabinose-inducible expression system, see Materials and methods |

*Appendix 2 Continued*

| Reagent type (species) or resource | Designation | Source or reference | Identifiers | Additional information |
|---|---|---|---|---|
| Recombinant DNA reagent | pUC18-Tn7t-pBAD-araE-pa2541 (plasmid) | This study | | Arabinose-inducible expression system, see Materials and methods |
| Recombinant DNA reagent | pUC18-Tn7t-pBAD-araE-pa4323 (plasmid) | This study | | Arabinose-inducible expression system, see Materials and methods |
| Recombinant DNA reagent | pUC18-Tn7t-pBAD-araE-*pa5114* (plasmid) | This study | | Arabinose-inducible expression system, see Materials and methods |
| Recombinant DNA reagent | pUC18-Tn7t-pBAD-araE-*pfl_rs01995* (plasmid) | This study | | Arabinose-inducible expression system, see Materials and methods |
| Recombinant DNA reagent | pUC18-Tn7t-pBAD-araE-*i35_rs27920* (plasmid) | This study | | Arabinose-inducible expression system, see Materials and methods |
| Recombinant DNA reagent | pUC18-Tn7t-pBAD-araE-*pspto_rs26695* (plasmid) | This study | | Arabinose-inducible expression system, see Materials and methods |
| Recombinant DNA reagent | pUC18-Tn7t-pBAD-araE-*ta05_rs00690* (plasmid) | This study | | Arabinose-inducible expression system, see Materials and methods |
| Recombinant DNA reagent | pTNS3 (plasmid) | *Choi et al., 2008* | | |
| Recombinant DNA reagent | pJRC115-*BTH_II0090*$^{S264A}$ (plasmid) | This study | | Point mutation construct, see Materials and methods |
| Recombinant DNA reagent | pJRC115-*BTH_I3226*$^{S264A}$ (plasmid) | This study | | Point mutation construct, see Materials and methods |
| Recombinant DNA reagent | pJRC115-*BTH_I2698*$^{S267A}$ (plasmid) | This study | | Point mutation construct, see Materials and methods |
| Recombinant DNA reagent | pJRC115-*BTH_I2701*$^{S267A}$ (plasmid) | This study | | Point mutation construct, see Materials and methods |
| Recombinant DNA reagent | pJRC115-Δ*BTH_I2691* (plasmid) | This study | | Deletion construct, see Materials and methods |
| Recombinant DNA reagent | pJRC115-Δ*BTH_I3225-8* (plasmid) | This study | | Deletion construct, see Materials and methods |
| Recombinant DNA reagent | pJRC115-Δ*BTH_II0089-94* (plasmid) | This study | | Deletion construct, see Materials and methods |

*Appendix 2 Continued on next page*

*Appendix 2 Continued*

| Reagent type (species) or resource | Designation | Source or reference | Identifiers | Additional information |
|---|---|---|---|---|
| Recombinant DNA reagent | pUC18T-miniTn7T-Tp-PS12-mCherry (plasmid) | *LeRoux et al., 2012* | | Construct for mCherry expression |
| Software, algorithm | Seqmagick | *Matsen Group, 2020*, https://github.com/fhcrc/seqmagick | | |
| Software, algorithm | TRANSIT TPP | https://transit.readthedocs.io/en/latest/tpp.html | | |
| Antibody | Anti-VSV-G (rabbit polyclonal) | MilliporeSigma | Cat# V4888-200UG | Western blot (1:5000 dilution) |
| Antibody | Anti-rabbit IgG HRP conjugated (goat) | MilliporeSigma | Cat# A6154-1ML | Western blot (1:5000 dilution) |
| Antibody | Anti-ribosome polymerase β (mouse monoclonal) | BioLegend | Cat# 663903; RRID: AB_2564524 | Western blot (1:1000 dilution) |
| Antibody | Anti-mouse IgG HRP conjugated (sheep) | MilliporeSigma | Cat# AC111P | Western blot (1:4000 dilution) |

## Appendix 3

**Appendix 3—table 1.** Oligonucleotides used in this study.

| Oligonucleotides 5′–3′ | Source |
|---|---|
| pEXG2_ΔPA2541_F1<br>CAAGCTTCTGCAGGTCGACTCTAGAAGGTGGAACCGGACCTGAAG | Integrated DNA Technology |
| pEXG2_ΔPA2541_R1<br>CTCAGGCCTGGGAAATCATGTCAGCCAGTCC | Integrated DNA Technology |
| pEXG2_ΔPA2541_F2<br>CATGATTTCCCAGGCCTGAGAGGGAACG | Integrated DNA Technology |
| pEXG2_ΔPA2541_R2<br>TAAGGTACCGAATTCGAGCTCGTATGACCCAGGCGGTCG | Integrated DNA Technology |
| pEXG2_ΔPA4323_F1<br>AGCTCGAGCCCGGGGATCCTCTAGAGACGCAGAACCCGATCGAG | Integrated DNA Technology |
| pEXG2_ΔPA4323_R1<br>AAAGCACGCCCGATGGCTTCATACGCGG | Integrated DNA Technology |
| pEXG2_ΔPA4323_F2<br>GAAGCCATCGGGCGTGCTTTGACGGATC | Integrated DNA Technology |
| pEXG2_ΔPA4323_R2<br>GGAAGCATAAATGTAAAGCAAGCTTGAGCGATGCCGAGCTGGA | Integrated DNA Technology |
| pEXG2_ΔPA5114_F1<br>CAAGCTTCTGCAGGTCGACTCTAGAGCCGCATGCCCTTGACCT | Integrated DNA Technology |
| pEXG2_ΔPA5114_R1<br>CGACCTCGGCAATCCATTGCATGCGTCGAATC | Integrated DNA Technology |
| pEXG2_ΔPA5114_F2<br>GCAATGGATTGCCGAGGTCGCATCGGAG | Integrated DNA Technology |
| pEXG2_ΔPA5114_R2<br>AATTCGAGCTCACCAGATCATCGGCGCCG | Integrated DNA Technology |
| pEXG2_ΔPA2536_F1<br>CAAGCTTCTGCAGGTCGACTCTAGACTCGGCCTGGCCCGAGC | Integrated DNA Technology |
| pEXG2_ΔPA2536_R1<br>CAGGCGAACCTCAGGGCGTTTCGTTCATCTCAGGCCTCC | Integrated DNA Technology |
| pEXG2_ΔPA2536_F2<br>GGAGGCCTGAGATGAACGAAACGCCCTGAGGTTCGCCTG | Integrated DNA Technology |
| pEXG2_ΔPA2536_R2<br>TAAGGTACCGAATTCGAGCTCCACATCGGTCTCAGCGAAGC | Integrated DNA Technology |
| pEXG2_ΔPA4320_F1<br>AGCTCGAGCCCGGGGATCCTCTAGATGTTCGGCTTCTACATCATGAAC | Integrated DNA Technology |

*Appendix 3—table 1 Continued on next page*

*Appendix 3—table 1 Continued*

| Oligonucleotides 5′–3′ | Source |
|---|---|
| pEXG2_ΔPA4320_R1<br>TGCGCCAGCCCACGCTGGCGTCAGTCAG | Integrated DNA Technology |
| pEXG2_ΔPA4320_F2<br>CGCCAGCGTGGGCTGGCGCAGCCTTTTC | Integrated DNA Technology |
| pEXG2_ΔPA4320_R2<br>GGAAGCATAAATGTAAAGCAAGCTTGTCGGCGGTTTCCTCGCT | Integrated DNA Technology |
| pEXG2_ΔPA5114/PA5113_F1<br>CAAGCTTCTGCAGGTCGACTCTAGAGCCGCATGCCCTTGACCT | Integrated DNA Technology |
| pEXG2_ΔPA5114/PA5113_R1<br>TGTTCGGCGAAATCCATTGCATGCGTCGAATC | Integrated DNA Technology |
| pEXG2_ΔPA5114/PA5113_F2<br>GCAATGGATTTCGCCGAACAAGCCATGAG | Integrated DNA Technology |
| pEXG2_ΔPA5114/PA5113_R2<br>TAAGGTACCGAATTCGAGCTCAACAGCCAGACCACGATGTAGC | Integrated DNA Technology |
| pEXG2_ΔPA5113_F1<br>CAAGCTTCTGCAGGTCGACTCTAGATTGCTAGGGGTGCTGGCG | Integrated DNA Technology |
| pEXG2_ΔPA5113_R1<br>ATGGCTTGTTGGAGCGCGTCATGGCTGC | Integrated DNA Technology |
| pEXG2_ΔPA5113_F2<br>GACGCGCTCCAACAAGCCATGAGCCGGTTC | Integrated DNA Technology |
| pEXG2_ΔPA5113_R2<br>TAAGGTACCGAATTCGAGCTCAACAGCCAGACCACGATGTAG | Integrated DNA Technology |
| pEXG2_ΔPA5112_F1<br>CAAGCTTCTGCAGGTCGACTCTAGATACCGCTGGCAGTTGCCG | Integrated DNA Technology |
| pEXG2_ΔPA5112_R1<br>AGAAGTCCAGCGCCATTCTGATCATTCTCTTACTC | Integrated DNA Technology |
| pEXG2_ΔPA5112_F2<br>CAGAATGGCGCTGGACTTCTGAAACGGCGGC | Integrated DNA Technology |
| pEXG2_ΔPA5112_R2<br>TAAGGTACCGAATTCGAGCTCCGCGCAACCGCCGGTTGG | Integrated DNA Technology |
| pEXG2_ΔPA3267_F1<br>CAAGCTTCTGCAGGTCGACTCTAGATCTACATCGACTTCGACG | Integrated DNA Technology |
| pEXG2_ΔPA3267_R1<br>ATCAGCGAGCTTGGGTCATCGTCCTTGTTAC | Integrated DNA Technology |
| pEXG2_ΔPA3267_F2<br>GATGACCCAAGCTCGCTGATCGATCCGC | Integrated DNA Technology |

*Appendix 3—table 1 Continued on next page*

*Appendix 3—table 1 Continued*

| Oligonucleotides 5′–3′ | Source |
|---|---|
| pEXG2_ΔPA3267_R2<br>TAAGGTACCGAATTCGAGCTCTTCGCCGGCCTGTTCGAAG | Integrated DNA Technology |
| pEXG2_ΔPFL_RS01995_F1<br>CAAGCTTCTGCAGGTCGACTCTAGAAACTACAACGTCAGCCTG | Integrated DNA Technology |
| pEXG2_ΔPFL_RS01995_R1<br>TTCATTCAACGGTTTGCCTATCCATTGCATGTGTCG | Integrated DNA Technology |
| pEXG2_ΔPFL_RS01995_F2<br>CGACACATGCAATGGATAGGCAAACCGTTGAATGAA | Integrated DNA Technology |
| pEXG2_ΔPFL_RS01995_R2<br>TAAGGTACCGAATTCGAGCTCCTGCGACCACACCAGCG | Integrated DNA Technology |
| pEXG2_PA2541_VSVG_F1<br>CAAGCTTCTGCAGGTCGACTCTAGAGCGCTGATACGGACCATGC | Integrated DNA Technology |
| pEXG2_PA2541_VSVG_R1<br>TTTTCCTAATCTATTCATTTCAATATCTGTATAGGCCTGGCCGTCGCTGCCCCG | Integrated DNA Technology |
| pEXG2_PA2541_VSVG_F2<br>TATACAGATATTGAAATGAATAGATTAGGAAAATGAGAGGGAACGGGCGAAC | Integrated DNA Technology |
| pEXG2_PA2541_VSVG_R2<br>TAAGGTACCGAATTCGAGCTCCTCCTGCGGCGCACGATG | Integrated DNA Technology |
| pEXG2_PA4323_VSVG_F1<br>CAAGCTTCTGCAGGTCGACTCTAGATGGAGTTCCACCAGTTGCGCG | Integrated DNA Technology |
| pEXG2_PA4323_VSVG_R1<br>TGATCCGTCATTTTCCTAATCTATTCATTTCAATATCTGTATAAAGCACGCCGGCGCGCT | Integrated DNA Technology |
| pEXG2_PA4323_VSVG_F2<br>TATACAGATATTGAAATGAATAGATTAGGAAAATGACGGATCAGACCGACTG | Integrated DNA Technology |
| pEXG2_PA4323_VSVG_R2<br>TAAGGTACCGAATTCGAGCTCAAGGTACACTTCTCCGCC | Integrated DNA Technology |
| pEXG2_PA5114_VSVG_F1<br>CAAGCTTCTGCAGGTCGACTCTAGACTGGAGCTGCGCTACCTGTTCG | Integrated DNA Technology |
| pEXG2_PA5114_VSVG_R1<br>TAATCTATTCATTTCAATATCTGTATATGGCTGCTCGGCCTCCGA | Integrated DNA Technology |
| pEXG2_PA5114_VSVG_F2<br>GATATTGAAATGAATAGATTAGGAAAATCGGAGGCCGAGCAGCCA | Integrated DNA Technology |
| pEXG2_PA5114_VSVG_R2<br>TAAGGTACCGAATTCGAGCTCGCTCGGACCAGATCATCGGC | Integrated DNA Technology |
| pEXG2_PA5113_VSVG_F1<br>CAAGCTTCTGCAGGTCGACTCTAGACTGCGTCTGGCCTGGCCG | Integrated DNA Technology |

*Appendix 3—table 1 Continued on next page*

*Appendix 3—table 1 Continued*

| Oligonucleotides 5′–3′ | Source |
|---|---|
| pEXG2_PA5113_VSVG_R1<br>TTTTCCTAATCTATTCATTTCAATATCTGTATATGGCTTGTTCGGCGAGGAAC | Integrated DNA Technology |
| pEXG2_PA5113_VSVG_F2<br>TATACAGATATTGAAATGAATAGATTAGGAAAATGAGCCGGTTCCGCGCTATG | Integrated DNA Technology |
| pEXG2_PA5113_VSVG_R1<br>TAAGGTACCGAATTCGAGCTCGTCGGGCAACAGCCAGAC | Integrated DNA Technology |
| pUC18_PA2541_F<br>AGC GAATTCGAGCTCGGTACCACGGGAGGAAAG ATGATTTCCGTCTATCAACTC | Integrated DNA Technology |
| pUC18_PA2541_R<br>CTCATCCGCCAAAACAGCCAAGCTTTCAGGCCTGGCCGTCGC | Integrated DNA Technology |
| pUC18_PA4323_F<br>AGCGAATTCGAGCTCGGTACCACGGGAGGAAAGATGAAGCCATCGCGCGCCCTGCTGG | Integrated DNA Technology |
| pUC18_PA4323_R<br>CTCATCCGCCAAAACAGCCAAGCTTTCAAAGCACGCCGGCGCGCTTCCAG | Integrated DNA Technology |
| pUC18_PA5114_F<br>AGCGAATTCGAGCTCGGTACCACGGGAGGAAAGATGCAATGGATTTTCATGCTGG | Integrated DNA Technology |
| pUC18_PA5114_R<br>CTCATCCGCCAAAACAGCCAAGCTTTCATGGCTGCTCGGCCTC | Integrated DNA Technology |
| pUC18_*pfl_rs01995*_F<br>AGCGAATTCGAGCTCGGTACCACGGGAGGAAAGATGCAATGGATATTCATGCTGC | Integrated DNA Technology |
| pUC18_*pfl_rs01995*_R<br>CTCATCCGCCAAAACAGCCAAGCTTTCACGATGACACTCCTTCATTC | Integrated DNA Technology |
| pUC18_*i35_rs27920*_F<br>AGCGAATTCGAGCTCGGTACCACGGGAGGAAAGATGAACTGGGCATTCGCCG | Integrated DNA Technology |
| pUC18_*i35_rs27920*_R<br>CTCATCCGCCAAAACAGCCAAGCTTTCATGGCTGGCCGTCCTG | Integrated DNA Technology |
| pUC18_*pspto_rs26695*_F<br>AGCGAATTCGAGCTCGGTACCACGGGAGGAAAGATGCTTTGGATTTGTCTGGTAG | Integrated DNA Technology |
| pUC18_*pspto_rs26695*_R<br>CTCATCCGCCAAAACAGCCAAGCTTTCATGGCGTTTCAGGCGCTG | Integrated DNA Technology |
| pUC18_*ta05_rs00690*_F<br>AGCGAATTCGAGCTCGGTACCACGGGAGGAAAGATGGACGACCTTTTAATCCTG | Integrated DNA Technology |
| pUC18_*ta05_rs00690*_R<br>CTCATCCGCCAAAACAGCCAAGCTTTCATTTGTTTTCTCCAGCTTTG | Integrated DNA Technology |
| pJRC115_ *BTH_II0090*[S264A]_F1<br>TAAAACGACGGCCAGTGCCAAGCTTCGACGGATCAGCGTTTCAAGCTGC | Integrated DNA Technology |

*Appendix 3—table 1 Continued on next page*

*Appendix 3—table 1 Continued*

| Oligonucleotides 5′–3′ | Source |
|---|---|
| pJRC115_ *BTH_II0090*$^{S264A}$_R1<br>ACCTTGGGCATGACCCATCACCGTGATCGTTTCGTG | Integrated DNA Technology |
| pJRC115_ *BTH_II0090*$^{S264A}$_F2<br>GGTCATGCCCAAGGTACGATCATCACGCTGCTCG | Integrated DNA Technology |
| pJRC115_ *BTH_II0090*$^{S264A}$_R2<br>GCTCGGTACCCGGGGATCCTCTAGACGGCTCGGCACGATGCGC | Integrated DNA Technology |
| pJRC115_ *BTH_I3226*$^{S264A}$_F1<br>TAAAACGACGGCCAGTGCCAAGCTTCGACGGATCAGCGTTTCAAGCTGC | Integrated DNA Technology |
| pJRC115_ *BTH_I3226*$^{S264A}$_R1<br>ACCTTGGGCATGACCCATCACCGTGATCGTTTCGTG | Integrated DNA Technology |
| pJRC115_ *BTH_I3226*$^{S264A}$_F2<br>GGTCATGCCCAAGGTACGATCATCACGCTGCTCG | Integrated DNA Technology |
| pJRC115_ *BTH_I3226*$^{S264A}$_R2<br>GCTCGGTACCCGGGGATCCTCTAGACGGCTCGGCACGATGCGC | Integrated DNA Technology |
| pJRC115_ *BTH_I2698*$^{S267A}$_F1<br>TCAATCAGTATCTAGAGGGACACCTTTCTCAAGCGAAATC | Integrated DNA Technology |
| pJRC115_ *BTH_I2698*$^{S267A}$_R1<br>GCCGCGCGCGAAACCAAACACATAAAGGCGAATGCG | Integrated DNA Technology |
| pJRC115_ *BTH_I2698*$^{S267A}$_F2<br>GGTTTCGCGCGCGGCGCAGCGGAAGCTCGCACGTTTTC | Integrated DNA Technology |
| pJRC115_ *BTH_I2698*$^{S267A}$_R2<br>TGTTAAGCTAGAATTCCATCAAACCCGCTGTCCCATGCTC | Integrated DNA Technology |
| pJRC115_ *BTH_I2701*$^{S267A}$_F1<br>TAAAACGACGGCCAGTGCCAAGCTTGAAAGACGAGCAGGACGCG | Integrated DNA Technology |
| pJRC115_ *BTH_I2701*$^{S267A}$_R1<br>ACCCCGCGCGAAACCGAATACGTATAGCCGAATGCG | Integrated DNA Technology |
| pJRC115_ *BTH_I2701*$^{S267A}$_F2<br>GGTTTCGCGCGGGGTGCAGCGGAGGCTCGCAC | Integrated DNA Technology |
| pJRC115_ *BTH_I2701*$^{S267A}$_R2<br>GCTCGGTACCCGGGGATCCTCTAGACTCGACCTCCAGCAGATCG | Integrated DNA Technology |
| pJRC115_ Δ*BTH_I2691*_F1<br>TAAAACGACGGCCAGTGCCAAGCTTATTTCAAGCGCGGCCAGTC | Integrated DNA Technology |
| pJRC115_ Δ*BTH_I2691*_R1<br>ATCTATGCGAGCTTCCCGCCATTTTTATTCC | Integrated DNA Technology |
| pJRC115_ Δ*BTH_I2691*_F2<br>GGCGGGAAGCTCGCATAGATGAGTGATG | Integrated DNA Technology |

*Appendix 3—table 1 Continued on next page*

*Appendix 3—table 1 Continued*

| Oligonucleotides 5′–3′ | Source |
| --- | --- |
| pJRC115_ ΔBTH_I2691_R2<br>GCTCGGTACCCGGGGATCCTCTAGATTCTGTCAATACTTAAAATACAATTTTC | Integrated DNA Technology |
| pJRC115_ ΔBTH_II0089-94_F1<br>TAAAACGACGGCCAGTGCCAAGCTTGAATCAGTGCATCGCTGTAC | Integrated DNA Technology |
| pJRC115_ ΔBTH_II0089-94_R1<br>ATGTTTTGCTTAGCGTATCGTTCAAATTGG | Integrated DNA Technology |
| pJRC115_ ΔBTH_II0089-94_F2<br>CGATACGCTAAGCAAAACATGAGATTATTGAAGAG | Integrated DNA Technology |
| pJRC115_ ΔBTH_II0089-94_R2<br>GCTCGGTACCCGGGGATCCTCTAGATATGCAACGCATTGCCGAAAC | Integrated DNA Technology |
| pJRC115_ ΔBTH_I3225-8_F1<br>TAAAACGACGGCCAGTGCCAAGCTTCCGGTCAATATACCACCATC | Integrated DNA Technology |
| pJRC115_ ΔBTH_I3225-8_R1<br>ATGTGCGCCCCGTATCGTTCAAATTGGTCAC | Integrated DNA Technology |
| pJRC115_ ΔBTH_I3225-8_F2<br>GAACGATACGGGGCGCACATAATAAGACTTG | Integrated DNA Technology |
| pJRC115_ ΔBTH_I3225-8_R2<br>GCTCGGTACCCGGGGATCCTCTAGAATTTGCTGTTTCTGCTGATG | Integrated DNA Technology |
| PCR_1A<br>GTCTCGTGGGCTCGGAGATGTGTATAAGAGACAGGGGGGGGGGGGGGGGGG | Integrated DNA Technology |
| PCR_1B<br>TCATCGGCTCGTATAATGTGTGG | Integrated DNA Technology |
| PCR_2A<br>CAAGCAGAAGACGGCATACGAGATTCGCCTTAGTCTCGTGGGCTCGG | Integrated DNA Technology |
| PCR_2B<br>CAAGCAGAAGACGGCATACGAGATCTAGTACGGTCTCGTGGGCTCGG | Integrated DNA Technology |
| PCR_2C<br>AATGATACGGCGACCACCGAGATCTACACCTAGAGACCGGGGACTTATCAGCCAACCTGTTA | Integrated DNA Technology |
| Seq_primer<br>CTAGAGACCGGGGACTTATCAGCCAAC | Integrated DNA Technology |

