## [Editor Report]

This study identifies novel (Arc) pathways that mediate resistance of bacteria to attacks by other bacteria. The identified genes appear to specifically protect host cells from the action of bacterial toxins. In particular, one system (Arc3) is shown to protect *Pseudomonas aeruginosa* from Type-6 secretion phospholipase effectors. Thus, while it is growing increasingly clear that bacteria have evolved numerous protection mechanisms against phages, the study expands this concept to bacteria.

---

## [Decision Letter]

**Decision letter after peer review:**

Thank you for submitting your article "Coordinately regulated interbacterial antagonism defense pathways constitute a bacterial innate immune system" for consideration by *eLife*. Your article has been reviewed by 2 peer reviewers, and the evaluation has been overseen by a Reviewing Editor and Wendy Garrett as the Senior Editor. The following individuals involved in review of your submission have agreed to reveal their identity: Lotte Sogaard-Andersen (Reviewer #2).

The reviewers have discussed their reviews with one another, and the Reviewing Editor with input from the senior editor. Consensus was that the manuscript is of significant import and interest but will require revision. We have drafted this document to help you prepare a revised submission.

Essential revisions:

1) The authors define the arc1-arc3 genes as part of a "bacterial innate immune system". This should be toned down, because it is not proven that they allow self- vs non-self-discrimination. In fact, the heterologous expression of Arc3B protects as well in Pa (presumably even Arc3B encoded in a B. thai) which argues against true self -vs non-self-discrimination.

2) The protective function of Arc3B could be further clarified: does it act directly on Tle3 or indirectly via a converging pathway that removes LPE or MLCL or acts on them? While it is understood that defining the exact molecular function of Arc3B is beyond the immediate scope of this manuscript, it should be possible to discriminate, testing for example if Arc3B can can protect against another phospholipase or whether LPE/MCL levels can be affected by Arc3B independently of Tle3. And last, does Arc3B require Arc3A for activity?

3) The manuscript is densely written and should be edited to make it an easier read for a broad audience.

*Reviewer #1 (Recommendations for the authors):*

1. line 126: Arc1 inactivation is described as resulting in a modest reduction in B. thai survival. My reading of the figure is that Arc1 inactivation results in a modest increase in B. thai survival. Please clarify.

2. line 126-127: I do not understand these comparisons. Maybe explain in a bit more detail.

3. Figure 2B: What is plotted on the Y-axis?

*Reviewer #2 (Recommendations for the authors):*

The authors define the arc1-arc3 genes as part of a "bacterial innate immune system". I found this choice questionable since it is not proven that they allow self- vs non-self-discrimination. In fact, since heterologous expression of Arc3B can protect as well in Pa (presumably even Arc3B encoded in a B. thai), it's not a true self -vs non-self-discrimination. Would referring to it as an antitoxin not be suitable instead?

The underpinnings protective (antitoxin) effect of Arc3B remain murky: does it fulfill a structural role, perhaps sequestering Tle3 or interfering with its activity, stability or translocation? Or does it act via a converging pathway that removes LPE or MLCL or acts on them, in parallel to Aas? In the latter case, Arc3B should be possible to protect against phospholipid cleavage caused in other ways than Tle3-mediated intoxication. Can this be tested with another phospholipase for example (expressed in Pa for self-intoxication)? The authors have not shown that LPE and MLCL levels can be modulated by Arc3B (with or without Arc3A) in the absence of Tle3, begging the question whether Arc3B acts specifically on Tle3 or phospholipase-mediated damage in general. Conversely can Aas overexpression compensate for loss of Arc3B and/or Arc3A in response to Tle3 intoxication?

Were the Arc3B orthologs that were chosen for complementation only from species that encode Arc3A or also from those that encode a solitary Arc3B? This could be discussed.

Arc3B interacts with Arc3A, but it does not appear to have enzymatic activity. It would be useful to know if Arc3B requires Arc3A for activity (assessed by epistasis experiments using Arc3B/A single and double mutants). One would predict that Arc3B can act independently of Arc3A since Arc3A is not always co-conserved with Arc3B, in which case the association between Arc3A and Arc3B should not be important to protect against Tle3.

Line 111 and panel A with B of Figure1—figure supplement 3: I don't see the cumulative effect claimed when compared the triple mutant Δarc123 (panel A) to single deletions (panel B).

Figure 1—figure supplement 3A: the names of the deleted strains seem incorrect: Δarc1DΔarc1G should be Δarc2DΔarc2G; as well Δarc1AΔarc1B is Δarc3AΔarc3B.

Line 218: Protection by Arc3A is valid for Tle3 and not against Tle1 (that is also a phospholipase). Change "against interbacterial phospholipase toxins" to "against Tle3 interbacterial phospholipase toxin".

---

## [Author Response]

Essential revisions:1) The authors define the arc1-arc3 genes as part of a "bacterial innate immune system". This should be toned down, because it is not proven that they allow self- vs non-self-discrimination. In fact, the heterologous expression of Arc3B protects as well in Pa (presumably even Arc3B encoded in a B. thai) which argues against true self -vs non-self-discrimination.

We thank the reviewer(s) for raising this matter. One possibility here is that the reviewer(s) may misunderstand how producers of Tle3 (such as *B. thailandensis*) prevent self-intoxication. These organisms do not rely on Arc3A homologs to provide protection; indeed, *B. thai* lacks a discernable homolog of *arc3A* and we generally see no correlation between the presence of *arc3A* homologs and predicted phospholipase toxin genes across bacterial genomes*.* Rather, self-intoxication is prevented (with 100% efficacy) through the production of dedicated immunity proteins encoded adjacent to the effector genes.

Additionally, we note that in eukaryotic innate immune systems, discrimination between self and non-self typically involves a means of sensing the foreign invader coupled to induction of protective responses. In *P. aeruginosa,* production of Arc3A is controlled by the Gac/Rsm globally regulatory system, and we show that this is associated with low expression of Arc3A during pure culture growth (Figure 2—figure supplement 2). Previously, we demonstrated that the Gac/Rsm pathway is activated specifically by the presence of an antagonistic competitor, via the lysis of a subset of the population and release of an as yet unidentified cellular component that serves as the activating signal (LeRoux *et al.*, 2015. *eLife*). Thus, paralleling eukaryotic innate immune system effectors, Arc3A becomes activated only under conditions where a foreign threat is perceived.

Finally, we would respectfully disagree with the assertion that self-vs. non-self discrimination is the sole function of an immune system. A key function of the eukaryotic innate immune system is the detection of and response to damaged cells (Kono and Rock, 2008. *Nat. Rev. Microbiol.*). This occurs through the detection of so-called danger associated molecular patterns (DAMPs), which can be generated either through the action of foreign invaders or self-derived sources of cellular damage (Shi *et al.,* 2003 *Nature;* Martinon *et al.*, 2006. *Nature*).

However, these considerations aside, at the reviewers’ request, we have toned down the language we use to describe the pathways we have discovered in both the discussion (p15, line 390) and the title of the paper (now modified to “Discovery of coordinately regulated pathways that provide innate protection against interbacterial antagonism”).

2) The protective function of Arc3B could be further clarified: does it act directly on Tle3 or indirectly via a converging pathway that removes LPE or MLCL or acts on them? While it is understood that defining the exact molecular function of Arc3B is beyond the immediate scope of this manuscript, it should be possible to discriminate, testing for example if Arc3B can can protect against another phospholipase or whether LPE/MCL levels can be affected by Arc3B independently of Tle3. And last, does Arc3B require Arc3A for activity?

We agree that it would be of interest to further clarify the protective function of Arc3A. We have attempted numerous experiments in an attempt to determine whether Arc3A can provide protection against other phospholipases, but have been met with a number of technical challenges. *B. thailandensis* produces a second phospholipase effector, Tle1. This protein has a less significant impact on *P. aeruginosa* growth than Tle3, but by reducing *P. aeruginosa* defenses through inactivation of the T6SS in this organism, we were able to detect a small but significant level of inhibition attributable to Tle1 (Author response image 1) . Inactivation of Ar3A under these conditions did not further sensitize *P. aeruginosa* to the activity of Tle1. However, a key caveat of this experiment is that the impact of Arc3A on Tle1-mediated intoxication in these assays may be masked by the stronger effect of Tle3. A “publishable” experiment would involve inactivating Tle3 to isolate the effect of Tle1-mediated intoxication. However, two homologous but non-identical copies of both *tle3* and *tle1* are encoded in *B. thai*, and generating the quadruple mutant necessary to perform this experiment in a controlled fashion has proved technically challenging. A second avenue we have explored is to evaluate whether inactivation of Arc3A results in sensitivity to other antagonists that encode known or predicted phospholipase toxins. Inactivation of Arc3A failed to sensitize *P. aeruginosa* to intoxication by six different phospholipase-encoding strains from five species. However, we are not confident in the interpretation of this result because each of the toxin-producing strains employed is significantly out-competed by *P. aeruginosa*, even when the experiment is initiated at a high donor:recipient ratio (, see strongly negative C.I. values), which is perhaps not surprising given the extensive offensive and defense arsenal this organism can employ. Thus, we have been unable to conclusively evaluate whether Arc3A confers protection against additional lysophospholipid-generating toxins.

**Author response image 1. sa2fig1:** Inactivation of Arc3A does not detectably impact *P. aeruginosa* growth in competition with assorted phospholipase producing species. (**A**) Recovery of *P. aeruginosa* with the indicated genotypes following growth in competition with an excess of wild-type or Tle1-inactivated *B. thai.* (**B and C**) Recovery of *P. aeruginosa* (**B**) or competitive index obtained (**C**; final donor:recipicient ratio divided by initial donor:recipienct ratio) following growth in competition with the indicated organisms.

To evaluate whether Arc3A can affect LPE levels in the absence of Tle3, we have isolated spheroplasts of *P. aeruginosa* expressing Arc3A or a derivative in which *arc3A* is deleted and incubated these with radiolabeled LPE. We find that Arc3A does not significantly affect the rate of LPE degradation in this assay (Figure 4G). This result was included but not highlighted in the original manuscript. We have now added mention of this finding to the results (p. 14, lines 365-366).

Finally, while our Tn-seq results indicate that Arc3B contributes to *P. aeruginosa* competitiveness, in pairwise competition experiments we see no evidence that Arc3B is required for Arc3A activity. This finding is now incorporated into the revised manuscript (*revised* Figure 3—figure supplement 1 and p12, lines 267-276). Arc3B is encoded downstream of Arc3A, suggesting the difference in phenotype we observed between the two assays is not due merely to polar effects of transposon insertion. Of note, we also found that transposon insertions in *aas* were associated with reduced *P. aeruginosa* fitness under the conditions of our tn-seq screen, but similarly, deletion of this gene did not have an effect on competitiveness in pairwise competitions. These findings suggest the transposon screen represents a more sensitive assay for the detection of genes important for fitness during interbacterial competition, and thus that Arc3B plays an important but auxiliary role in *P. aeruginosa* defense against antagonism.

3) The manuscript is densely written and should be edited to make it an easier read for a broad audience.

We regret any difficulty that our writing style may have caused and we thank the reviewer for providing this comment. We have incorporated revisions throughout the manuscript to improve readability:

Reviewer #1 (Recommendations for the authors):1. line 126: Arc1 inactivation is described as resulting in a modest reduction in B. thai survival. My reading of the figure is that Arc1 inactivation results in a modest increase in B. thai survival. Please clarify.

The reviewer is correct in that inactivation of Arc1 causes a modest increase in *B. thai* survival, i.e. a modest decrease in the ability of *P. aeruginosa* ability to kill *B. thai.* We regret the confusion caused by our original statement and have revised for clarity (p8, lines 143-145).

2. Line 126-127: I do not understand these comparisons. Maybe explain in a bit more detail.

We have revised the description of these experiments to provide more clarity (p. 7, line 131-p. 8, line 145)

3. Figure 2B: What is plotted on the Y-axis?

The specific metric presented in this figure was calculated by dividing the *P. aeruginosa* CFUs obtained for the indicated mutant by those obtained for the wild-type strain during competition with the indicated *B. thai* strain, and then multiplying by 100 to obtain a percentage. We have revised the figure legend to make this more clear.

Reviewer #2 (Recommendations for the authors):The authors define the arc1-arc3 genes as part of a "bacterial innate immune system". I found this choice questionable since it is not proven that they allow self- vs non-self-discrimination. In fact, since heterologous expression of Arc3B can protect as well in Pa (presumably even Arc3B encoded in a B. thai), it's not a true self -vs non-self-discrimination. Would referring to it as an antitoxin not be suitable instead?

See the response to Essential Revision #1 above.

The underpinnings protective (antitoxin) effect of Arc3B remain murky: does it fulfill a structural role, perhaps sequestering Tle3 or interfering with its activity, stability or translocation? Or does it act via a converging pathway that removes LPE or MLCL or acts on them, in parallel to Aas? In the latter case, Arc3B should be possible to protect against phospholipid cleavage caused in other ways than Tle3-mediated intoxication. Can this be tested with another phospholipase for example (expressed in Pa for self-intoxication)? The authors have not shown that LPE and MLCL levels can be modulated by Arc3B (with or without Arc3A) in the absence of Tle3, begging the question whether Arc3B acts specifically on Tle3 or phospholipase-mediated damage in general. Conversely can Aas overexpression compensate for loss of Arc3B and/or Arc3A in response to Tle3 intoxication?

We agree with the reviewer that a mechanistic explanation for the protected effect of Arc3B remains unclear. From an evolutionary perspective, it seems unlikely that Arc3A acts specifically to provide defense against one phospholipase toxin produced by another species. However, for technical reasons discussed above (see Essential Revision #2), we have yet to identify other phospholipase-targeting toxins against which Arc3A conclusively provides protection. As requested by the reviewer, we have evaluated whether Aas over-expression can compensate for Arc3A inactivation. Our data show that Aas over-expression does not alter the compromised defense of ∆*arc3A* (*new* Figure 4—figure supplement 3 and p14, lines 362-364)**.**

Were the Arc3B orthologs that were chosen for complementation only from species that encode Arc3A or also from those that encode a solitary Arc3B? This could be discussed.

The Arc3A homologs we evaluated for their ability to complement the *arc3A* deletion in *P. aeruginosa* were all from organisms that also encoded a homolog of Arc3B. The organisms encoding unaccompanied Arc3A homologs are all distantly related to *P. aeruginosa* (Gram-positives, cyanobacteria, etc., see Figure 3B) and thus we reasoned that poor expression of the genes would confound the experiment.

Arc3B interacts with Arc3A, but it does not appear to have enzymatic activity. It would be useful to know if Arc3B requires Arc3A for activity (assessed by epistasis experiments using Arc3B/A single and double mutants). One would predict that Arc3B can act independently of Arc3A since Arc3A is not always co-conserved with Arc3B, in which case the association between Arc3A and Arc3B should not be important to protect against Tle3.

The prediction of the reviewer is correct, in that Arc3B is not required for Ar3A activity. In the revised manuscript, we include the results of the suggested experiment, which demonstrates that inactivation of Arc3B alone or in combination with Arc3A does not affect *P. aeruginosa* competitiveness in pairwise competition assays with *B. thailandensis* (*revised* Figure 3—figure supplement 1 and p12, lines 267-276)*.* However, we note that in our Tn-seq assays, *arc3A* was depleted more than 3-fold following competition with antagonistic *B. thailandensis*. Importantly, *arc3A* is encoded downstream of *arc3B*, thus eliminating the possibility that this phenotype results from polar effects of the transposon. Thus, it appears that Arc3A does contribute to competitiveness under some conditions, but is clearly not required for Arc3B function in *P. aeruginosa*.

Line 111 and panel A with B of Figure1—figure supplement 3: I don't see the cumulative effect claimed when compared the triple mutant Δarc123 (panel A) to single deletions (panel B).

The cumulative effect we are referring to here is apparent when comparing the parental final CFU counts to those of the individual mutants or the triple mutant for a given experiment. We observe a degree of variability in the growth yield of the parent strain competed against *B. thai* in different experiments. To highlight the difference we are referring to, we have now added the fold difference value for CFUs obtained for the mutants vs. the parent strain to each graph (revised Figure 1—figure supplement 3).

Figure 1—figure supplement 3A: the names of the deleted strains seem incorrect: Δarc1DΔarc1G should be Δarc2DΔarc2G; as well Δarc1AΔarc1B is Δarc3AΔarc3B.

The reviewer is correct and we have fixed this error in the revised manuscript.

Line 218: Protection by Arc3A is valid for Tle3 and not against Tle1 (that is also a phospholipase). Change "against interbacterial phospholipase toxins" to "against Tle3 interbacterial phospholipase toxin".

We appreciate the issue raised here. We have revised this sentence to indicate more clearly that we currently only have evidence for protection against one type of phospholipase toxin provided by Arc3A (p 12, line 266).